# Self-emergence of robust solitons in a microcavity

Maxwell Rowley[1], Pierre-Henry Hanzard[1], Antonio Cutrona[1,2], Hualong Bao[1], Sai T. Chu[3], Brent E. Little[4], Roberto Morandotti[5], David J. Moss[6], Gian-Luca Oppo[7], Juan Sebastian Totero Gongora[1,2], Marco Peccianti[1,2] & Alessia Pasquazi[1,2 ✉]

In many disciplines, states that emerge in open systems far from equilibrium are determined by a few global parameters[1,2]. These states can often mimic thermodynamic equilibrium, a classic example being the oscillation threshold of a laser[3] that resembles a phase transition in condensed matter. However, many classes of states cannot form spontaneously in dissipative systems, and this is the case for cavity solitons[2] that generally need to be induced by external perturbations, as in the case of optical memories[4,5]. In the past decade, these highly localized states have enabled important advancements in microresonator-based optical frequency combs[6,7]. However, the very advantages that make cavity solitons attractive for memories—their inability to form spontaneously from noise—have created fundamental challenges. As sources, microcombs require spontaneous and reliable initiation into a desired state that is intrinsically robust[8–20]. Here we show that the slow non-linearities of a free-running microresonator-filtered fibre laser[21] can transform temporal cavity solitons into the system's dominant attractor. This phenomenon leads to reliable self-starting oscillation of microcavity solitons that are naturally robust to perturbations, recovering spontaneously even after complete disruption. These emerge repeatably and controllably into a large region of the global system parameter space in which specific states, highly stable over long timeframes, can be achieved.

Dissipative solitons—self-confined pulses that balance non-linear phase shift with wave dispersion (or diffraction) in driven and lossy systems—are ubiquitous, with passive Kerr cavities and passively mode-locked lasers being prime examples in optics[2,22]. As the field has matured, understanding the physics that sustains these solitary waves in passive mode-locking has enabled the development of strategies to ensure the reliable initiation into pulses that are robust to perturbations—ultimately driving the advancement of modern ultrafast laser technology[2,22,23].

A similar scenario now confronts microresonator-based optical frequency combs, or microcombs, which have enabled notable breakthroughs in metrology, telecommunications, quantum science and many other areas[24–39]. A robust, repeatable approach for initiating and reliably maintaining the microcomb into the same type of soliton state, particularly the single-soliton state, is widely acknowledged as critical, with recent notable progress[8–20]. Nonetheless, it largely remains the main outstanding challenge confronting this field.

Microcombs are based on the physics of cavity solitons[2,5], which are localized pulses that leave a large portion of the cavity in a strictly stable, low-energy state. However, this very stability indicates that no cavity-soliton state can grow from noise. Hence, they can only appear if directly 'written' through a dynamic, often complex, perturbation of one of the system variables[5,40–42]. This procedure is not trivial to control directly in microcavities and, in turn, makes it extremely difficult to

achieve a single configuration of global parameters that allows the combination of initiation, selection and robust maintenance of a given soliton state. Most of all, after a 'disrupting' event in which an external perturbation destroys the desired soliton state, the system does not naturally self-recover to the original state.

One approach to turnkey operation is to refrain from imposing strict stability on the low-energy state, allowing for the evolution of a periodic waveform that eventually forms a succession of stable solitary peaks (soliton crystals)[43–45]. This type of starting procedure can be controlled by acting on the modulational instability of the background state in double cavity systems[46–48]. As such, turnkey microcombs[13] by self-injection locking have shown an operating start-up point for multi-soliton states. Alternatively, by introducing a periodic modulation of the refractive index in the microcavity[49], single solitons have been generated by scanning the driving pump into resonance without passing through the well-known chaotic state. Slow non-linearities, such as the photorefractive effect in lithium niobate[12] or Brillouin scattering[16,17], have also been exploited to move the microcomb into a soliton state.

Nevertheless, all of these schemes now require a specific system preconfiguration and the ability to execute a precise dynamical path towards initiating the desired soliton state. These strict and critical conditions—especially regarding the phase configuration—markedly

[1]Emergent Photonics (Epic) Laboratory, Department of Physics and Astronomy, University of Sussex, Falmer, UK. [2]Emergent Photonics Research Centre and Department of Physics, Loughborough University, Loughborough, UK. [3]Department of Physics, City University of Hong Kong, Tat Chee Avenue, Hong Kong, Hong Kong SAR, China. [4]State Key Laboratory of Transient Optics and Photonics, Xi'an Institute of Optics and Precision Mechanics, CAS, Xi'an, China. [5]INRS-EMT, Varennes, Québec, Canada. [6]Optical Sciences Centre, Swinburne University of Technology, Hawthorn, Victoria, Australia. [7]SUPA, Department of Physics, University of Strathclyde, Glasgow, UK. ✉e-mail: a.pasquazi@lboro.ac.uk

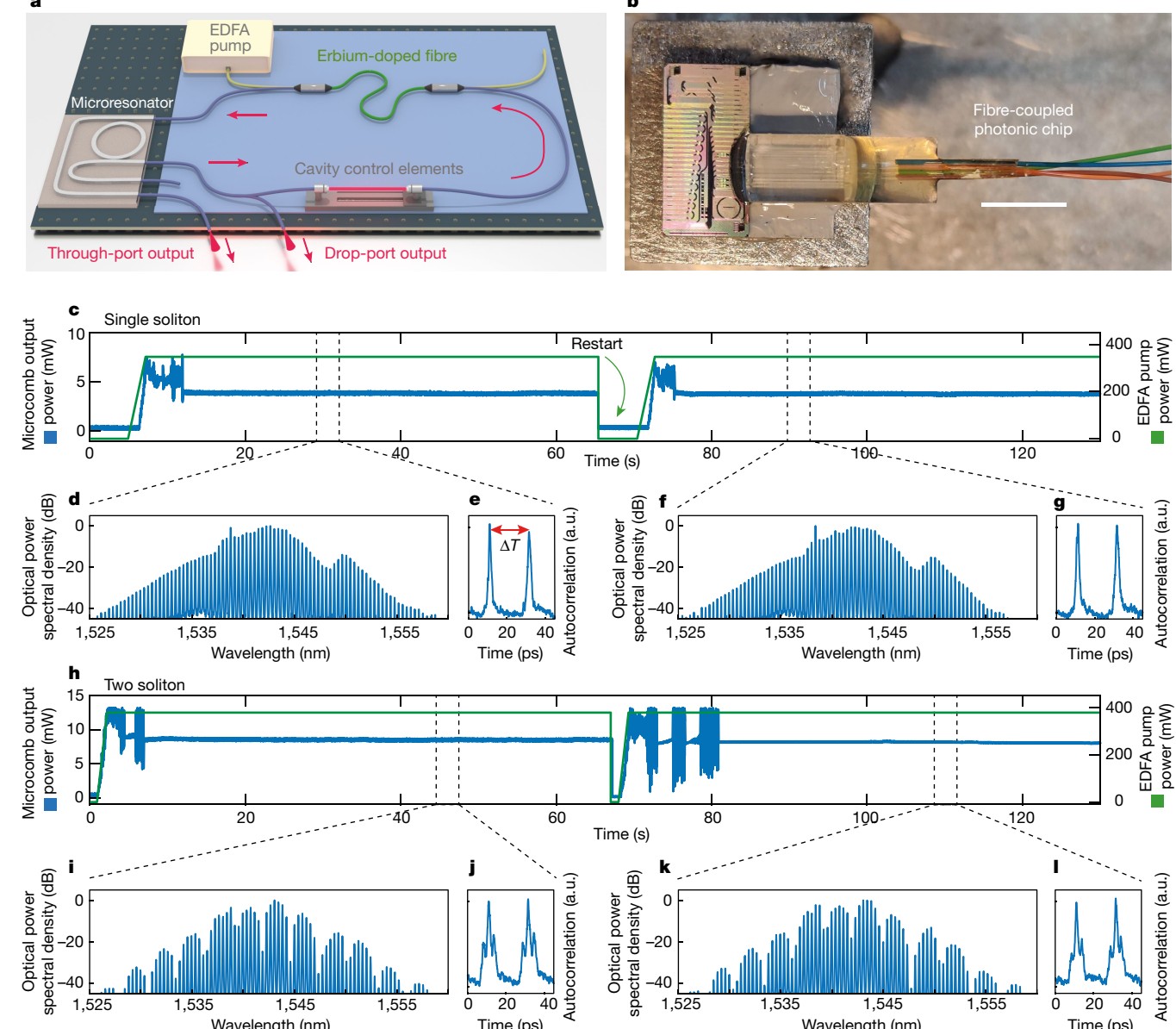

**Fig. 1 | Natural onset of cavity solitons. a**, Microcomb laser. The non-linear Kerr microresonator (FSR 48.9 GHz) completes the fibre laser cavity (FSR 95 MHz). The global cavity controls are highlighted: a section containing the variable EDFA 980 nm pump, an optical filter, polarization controls and a delay stage to roughly match the repetition rate of the fibre cavity with a submultiple of the microcavity FSR. The fibre-coupled output ports of the microresonator are highlighted. **b**, Picture of the microresonator photonic chip with integrated fibre coupling. Scale bar, 10 mm. **c**, Repeatable start-up of the same single-soliton state from the off state, a temporal measurement of the microcomb output power (blue line), stabilizing to 4 mW. The EDFA pump power (green line) is increased from 0 to 350 mW in 2 s. **d,e**, Output spectrum

(**d**) and autocorrelation (**e**) of the microcomb after the first start-up, at 30 s. $\Delta T = 20$ ps is the time period corresponding to one round-trip of the microcavity. **f,g**, Output spectrum (**f**) and autocorrelation (**g**) of the microcomb emitted after the second start-up, at 95 s. **h** Repeatable start-up of the same two-soliton state, selected by driving the EDFA at a higher regime power of 380 mW. A temporal measurement of the microcomb output power (blue line), stabilizing to 8 mW. The EDFA pump power (green line) is increased from 0 to 380 mW in 2 s. **i,j**, Spectrum (**i**) and autocorrelation (**j**) of the microcomb after the first start-up, at 45 s. **k,l**, Spectrum (**k**) and autocorrelation (**l**) after the second start-up, at 110 s.

increase the system's susceptibility to external perturbations and, most importantly, do not offer any pathway for the soliton states to spontaneously recover.

In this article, we introduce a fundamental approach to solving this challenge. Our strategy relies on judiciously tailoring a slow and energy-dependent non-linearity to transform a specifically targeted soliton state into the dominant attractor of the system. As a result, the chosen soliton state consistently appears simply by turning the system on and, just as notably, naturally recovers after drastic perturbations that entirely disrupt the solitons. This methodology allows the system to

persist in the same soliton state under free-running operation over arbitrarily long timeframes, without any external control. Specific states, including single-soliton states, can be reliably generated by choosing the correct parameters that only have to be set once ('set-and-forget').

Figure 1 shows a simple embodiment of this approach based on a microresonator-filtered fibre laser[21,50,51] (Fig. 1a). An integrated microring resonator (Fig. 1b, free-spectral range (FSR) of 49.8 GHz) is nested within an erbium-doped fibre amplifier (EDFA) lasing cavity. We use a four-port ring resonator, measuring the output at both the 'drop' and 'through' ports, with the corresponding spectral features discussed

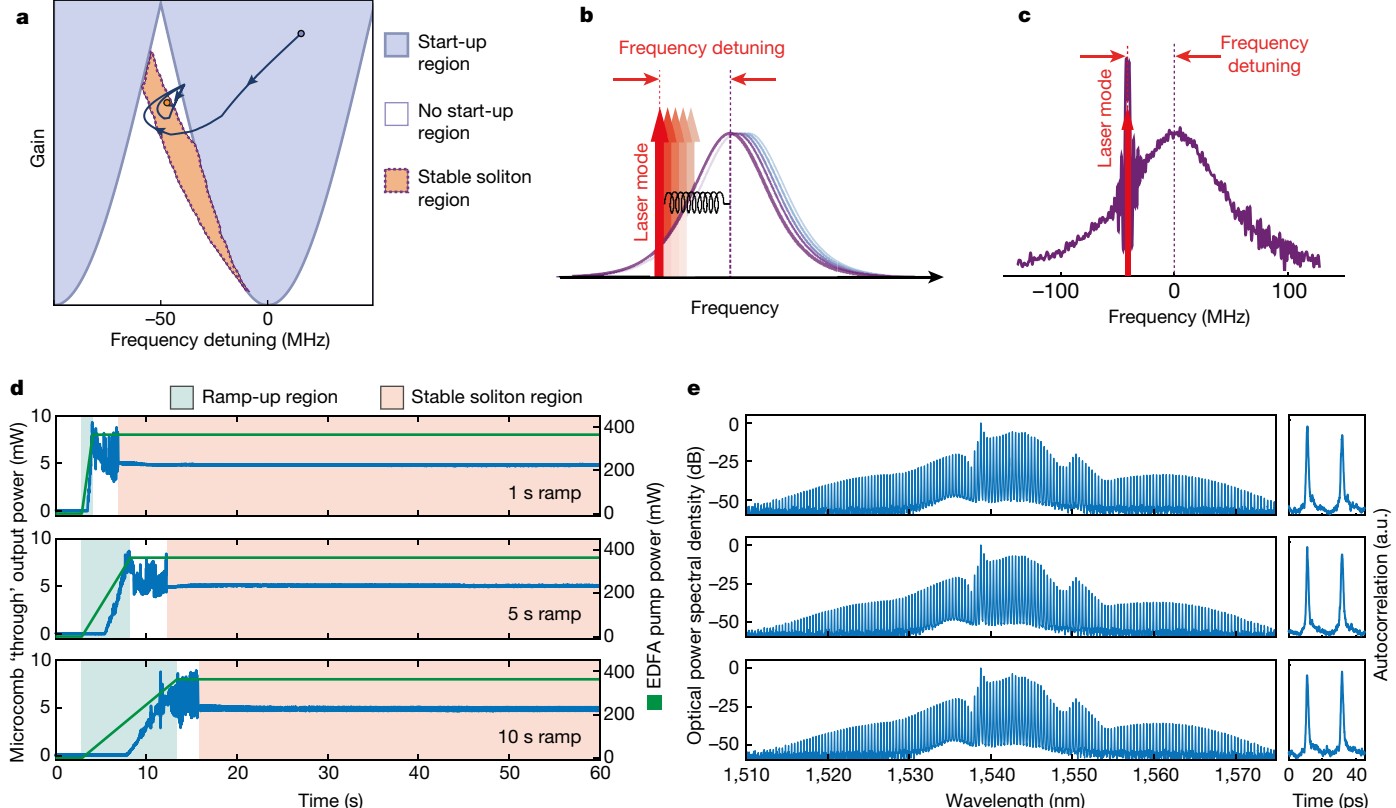

**Fig. 2 | Principle of operation for the natural onset and intrinsic stability of cavity solitons. a**, Diagram of states for a microcomb laser. Here the coordinates are two typical parameters, frequency detuning (*x* axis, here scaled to our experimental setting) and gain (*y* axis). The gain roughly correlates with the EDFA pump power; further details are in the Supplementary Information, Section S1. The start-up region is in blue. The stable solitons (orange) region is well within the no start-up (white) region. In our system, the soliton behaves as a dominant attractor (dark-blue path). Note that the regions with different soliton numbers are perfectly superimposed here; further details are in the Supplementary Information, Figs. 1 and 2. **b**, Microcavity resonance (purple) and laser modes (red) during stable soliton operation. The energy-dependent red-shift of the laser modes is greater than that of the

microcavity. As such, the system preferentially locks to the laser mode red-detuned to the microcavity resonance. The orange arrows highlight the frequency detuning parameter, defined as the difference between the microcavity central resonance and the laser mode. **c**, Laser scanning spectroscopy of a microcavity resonance (bandwidth 120 MHz, *Q* factor of $10^6$) under lasing conditions. The red-detuned lasing frequency is visible as a sharp peak highlighted by a red arrow. **d**, Experimental start-up of a single soliton from the off state. Microcomb output power versus time (blue) and EDFA pump power (green). The EDFA pump is ramped from 0 to 360 mW. The three panels indicate different ramp times of 1, 5 and 10 s, respectively. **e**, Experimental output spectra and autocorrelations (right inset) corresponding to the adjacent panels in **d**.

in Extended Data Fig. 1. Unless otherwise specified, we report data measured at the 'through' port. Here we use a roughly 2 m fibre loop with an optical path set to a multiple of the microcavity length in a tolerance of a few hundred micrometres (FSR 95 MHz). A 980-nm laser diode (EDFA pump) induces the optical gain in the amplifier. The system consistently and repeatably starts up into the same desired state by simply setting the EDFA pump power to a fixed value, as shown in Fig. 1c–l. We consistently achieve the same single-soliton state for an EDFA pump power of 350 mW. Figure 1c shows the microcomb output power, whereas Fig. 1d–g shows the corresponding spectra (Fig. 1d,f) and autocorrelation (Fig. 1e,g) examples for the intermediate and final states. Further, with the pump power set to 370 mW, the system consistently yields a two-soliton state (Fig. 1h–l).

Here we provide a simple description of the underlying phenomenon. We configure the microcomb with start-up parameters where the background state is unstable, thus allowing noise to grow and initiate the oscillation (Fig. 2a, blue region). Although this condition is usually incompatible with stable soliton states, in our case, two slow and energy-dependent non-linearities arising from the EDFA in the main fibre cavity, as well as the thermal response of the microresonator[52], non-locally modify the state of the system as the energy increases. This process intrinsically creates a dominant attractor: the system

moves from the laser start-up region into a distinct stability region for the desired soliton state, which is naturally formed and intrinsically maintained without any external control.

In general, the most challenging parameter to control for microcombs is the frequency position of the comb lines within the microcavity resonances, that is, the 'frequency detuning' parameter (Fig. 2a,b and Extended Data Figs. 2 and 3). We define this value as the average offset of the laser mode relative to the microcavity resonance centre that fundamentally defines the region of stability of the solitons. Controlling the detuning usually requires high accuracy in frequency, often translating into the need for strict start-up phase conditions[13] or turn-on procedures that are critically dependent on the ramp-up dynamics[8–16]. The key to our approach lies in controlling the magnitude of an effective non-local non-linearity that ultimately locks the comb lines into the desired position while maintaining the frequency detuning. The laser lines naturally follow the resonances when the system is perturbed, resulting in the critically important ability for the state to naturally reform, even after being entirely disrupted.

Although accurately controlling relevant changes in a non-linearity can be very challenging, we achieve it naturally by designing the double cavity to effectively balance the strong thermal non-linearity of the microresonator with the large non-linearity resulting from the EDFA.

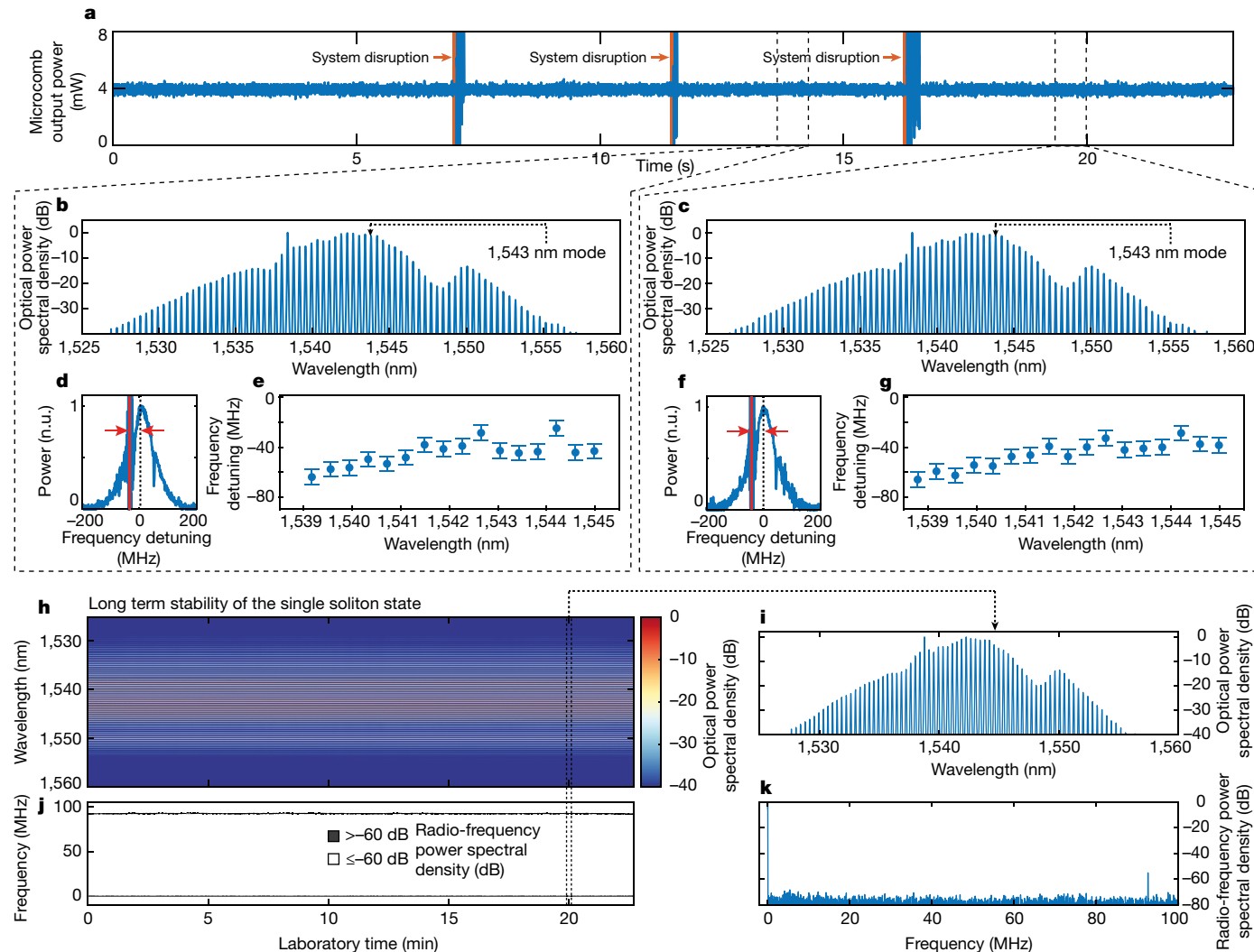

**Fig. 3 | 'Set-and-forget' operation, robust single soliton as a dominant attractor: recovery of the same single-soliton state from perturbations and long-term stability. a**, Output power of the single-soliton state. The system is perturbed three times with a mechanical disruption at roughly 7, 11 and 16 s (red arrow). **b,c**, Spectra of the system at 14.5 s (**b**) and 19.5 s (**c**), before and after the perturbation. **d**, Laser scanning spectroscopy of the 1,543 nm microcavity resonance under lasing condition. The frequency difference between the microcavity resonance centre (black dashed line) and the laser line (red line) defines the operative frequency detuning of the mode. **e**, Frequency detuning of a selection of microcomb lines extracted as in **d**.

**f,g**, Same as **d**,**e** at 19.5 s. **h**, Long-term robustness of the state, showing roughly half an hour of continuous operation of a single-soliton state. Temporal evolution of the measured optical spectrum. The colour bar shows the optical power spectral density. **i**, Typical optical spectrum from **h** (taken at 20 min). **j**, Evolution of the radio-frequency spectrum of **h** in time. Black and white colour map, with power spectral density above and below −60 dB reported in black and white, respectively. As expected, there is significant power spectral density only at zero frequency. **k**, Typical radio-frequency spectrum from **j**, at 20 min.

We do this by exploiting the small refractive index variation that results from changing the optical pump power of the gain material[53–55], which effectively controls this fundamental equilibrium. With the non-linear gain saturation, our control of the effective non-linearity enables selecting a specific state, including the soliton number. By acting on the global parameters of the system, including the EDFA pump power, the system consistently and robustly remains in the selected soliton state. The Supplementary Information, Section S2 and Extended Data include detailed theoretical modelling (Supplementary Information and Extended Data Fig. 2) and more extensive experimental results (Supplementary Information and Extended Data Figs. 4–8) to illustrate the system's fundamental physics comprehensively.

We do not observe any significant dependence of the operating state on the start-up dynamics of the EDFA pump: we simply need to turn the system on with fixed pump power. The value of this power needs to be determined only once (set-and-forget), which allows selecting,

for instance, single versus two-soliton states as in Fig. 1. In Fig. 2d, we repeatedly turned on the EDFA pump to 360 mW to obtain a broadband single soliton. Here, once the thermalization of the laser system is complete (here at roughly 5 s), the system consistently yields the same single-soliton state (Fig. 2e), regardless of the pump ramp-up time. The laser mode (Fig. 2c), measured using laser scanning spectroscopy, is red-detuned within the microcavity resonance, indicating that the system operates in its bistable region, where solitons both exist and are stable. Fundamentally, the soliton state does not require any specifically implemented 'writing procedure', which results in the system operating with exceptional robustness. Figure 3a shows that the soliton state consistently reappears even after strong system disruptions induced by external perturbations (Fig. 3a). The spectra in Fig. 3b,c show how the same soliton state reliably recovers and the comb lines return in the same position within the microcavity resonances, given our experimental accuracy (Fig. 3d–g). If left unperturbed, the

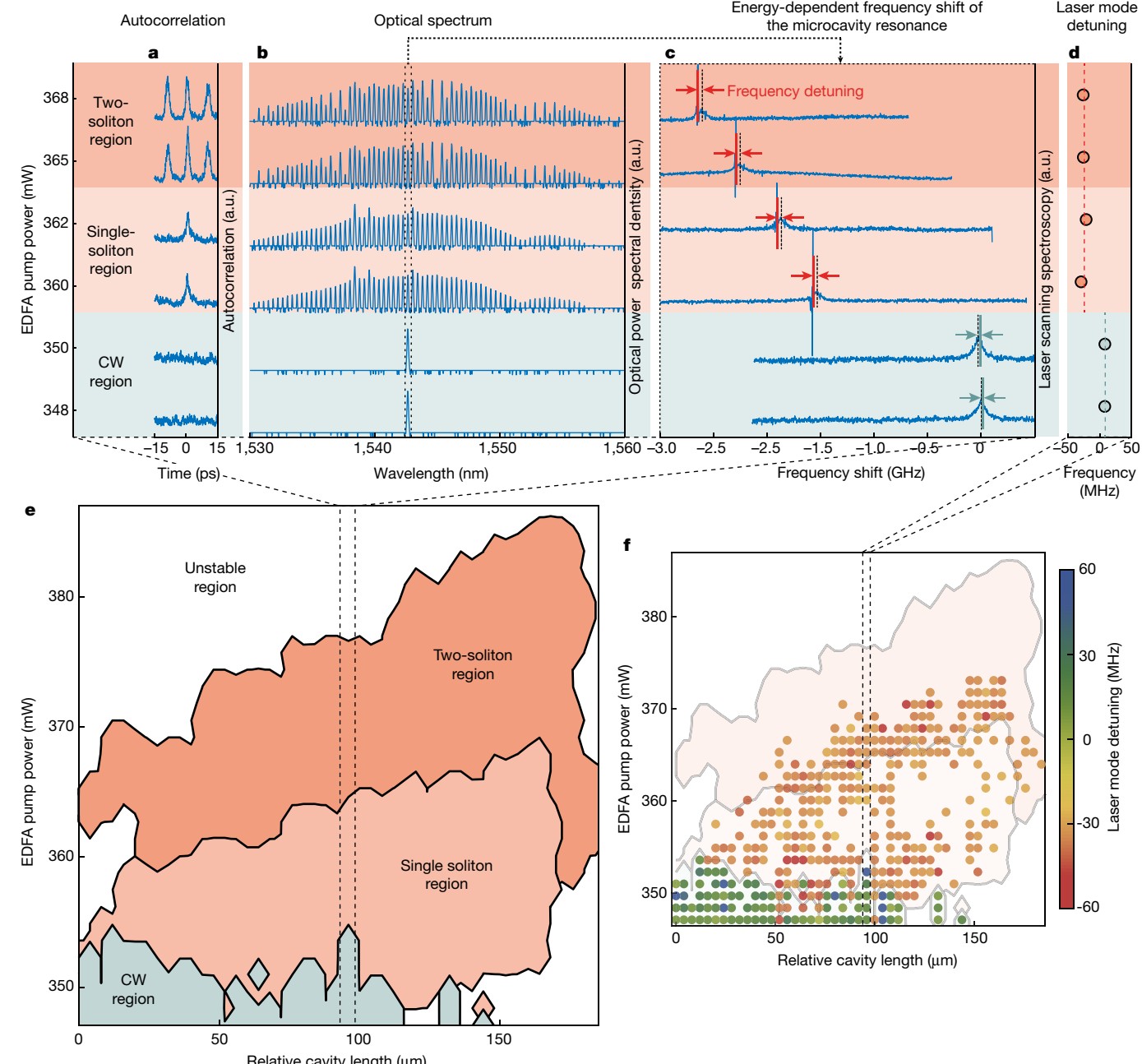

**Fig. 4 | Diagram of states. a–d,** System stationary states as a function of EDFA pump power, indicated in the *y* axis, highlighting the well-defined regions of existence for different types of state. Here these states consist of continuous wave (CW), single and two-soliton states. Stationary states are obtained by increasing the EDFA pump power from the laser threshold. **a,** Autocorrelation of typical states in the range. **b,** Optical spectrum. **c,** Laser scanning spectroscopy of the line at 1,543 nm, the arrows highlight the frequency detuning of the laser modes. The frequency shift of the modes is due to the thermal non-linearity of the microcavity. **d,** Average frequency detunings for all of the measured laser modes of the microcavity resonances, extracted as in **c** and measured with respect to the microcavity mode central frequency. **e,** Full diagram of states obtained over various cavity length settings in addition to the EDFA pump power. White regions are unstable states. Solitons are consistently found in a span of roughly 200 μm, indicating the independence of the system operation from the initial cavity phases of the system. **f,** Average frequency detuning of the laser mode extracted as in **d.** Distribution versus EDFA pump power and cavity length. Red to blue points indicate the frequency detuning from −60 to 60 MHz, as in the colour bar.

soliton state operates indefinitely. Figure 3h shows almost half an hour of continuous measurements of the same single-soliton spectrum, showing an ultra-low noise radio-frequency spectrum (Fig. 3j). Supplementary Information, Section S3 with Extended Data Figs. 9 and 10 studies the recovery dynamic of different states both experimentally and theoretically.

Figure 4 shows the system output state measured at the 'drop' port versus EDFA pump power, indicating that we consistently obtain continuous wave and single and two-soliton states, each in distinct ranges of the EDFA power. Laser scanning spectroscopy measurements of the microresonator resonance at 1,543 nm show a notable red-shift above 2 GHz, induced by the thermal non-linearity (Fig. 4c). This shift exceeds the main-cavity FSR (77 MHz) and the microcavity linewidth (150 MHz) by almost two orders of magnitude. Nonetheless, the soliton laser modes clearly lock to the red-detuned slope of the microcavity (Fig. 4d) in a small range of a few megahertz. Notably, the continuous

wave states are all locked onto the blue-detuned side of the microcavity, as is typical for these types of state. This clear locking phenomenon confirms the independence of the particular states from the position of the microcavity resonance. Furthermore, it highlights that the frequency detuning is a signature of the dominant attractor, determined solely by the selected EDFA pump power.

The experimental diagram of states shown in Fig. 4e (and Supplementary Information, Section S2) reflects this consistent behaviour even more strikingly. Figure 4e extends the measurements of Fig. 4a–d as a function of varying the laser cavity length, and Fig. 4f shows the corresponding average detunings of the states. Spanning the cavity length enables the systematic testing of the system's dependence on the initial cavity phase. The phase varies from 0 to 2π as the cavity length changes by an amount equal to the optical wavelength (1.5 μm). In general, even small phase variations (well below π) can strongly affect the type of soliton states obtained[13] or can even notably prevent the system from reaching any solitons state in the first place. In our system, conversely, we consistently and continuously obtain the same type of soliton state (for example, single soliton) even with cavity length variations that are hundreds of times larger than π, in the order of 200 μm. Such a large span clearly demonstrates that the formation of our soliton states is essentially cavity-phase independent.

In conclusion, we demonstrate the spontaneous initiation of cavity solitons, independent of any initial system conditions or detailed pump dynamics. These states are intrinsically stable and naturally self-recover after being disrupted. We achieve this by transforming the soliton states into dominant attractors of the system and experimentally demonstrating this approach in a microresonator-filtered fibre laser. This method is fundamental and very general, applicable to a wide range of systems, particularly those based on dual-cavity configurations such as self-injection locking[8,9,13]. Moreover, our theoretical model by using the very general Maxwell–Bloch equations shows that any common gain material can be used to tailor the non-local non-linearity. We measure a clear diagram of states as a function of two simple global system parameters—the EDFA pump power and laser cavity length—with large regions associated with the desired solitary states. More generally, in the field of pulsed lasers, this work provides an effective approach to achieving self-starting, broadband pulsed laser without fast saturable absorbers that are notoriously difficult to realize, particularly for ultrashort pulses[2,22]. Our work represents a key milestone in the development of microcombs, resulting in robust operation that naturally initiates and maintains cavity-soliton states, all of which are key requirements for real-world applications.

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

## Methods

### Setup

The experimental setup consists of nesting a high index doped silica[21,50,51], integrated ring resonator, with a FSR of roughly 48.9 GHz, a 1.3 million $Q$ factor and a positive (focusing) Kerr non-linear coefficient of about 200 times that of silica. The resonator is set in an add-drop configuration into an amplifying, polarization-maintaining, fibre cavity. Our samples use a fibre array glued directly on the chip (Fig. 1b), making the setup practical. Each coupling port has 1.5 dB losses leading to a total input-output coupling loss of 3 dB. Because the generated solitons have a very high conversion efficiency, the chip's input power was generally below 100 mW (Supplementary Information, Section S2); hence we operate well below the damage threshold of standard optical glue.

For Figs. 1–3, we used a fibre cavity with a FSR of roughly 95 MHz and a microcavity sample with linewidth <120 MHz. The results of Fig. 4 are for a longer cavity with a FSR roughly of 77 MHz and a microcavity sample with linewidth <150 MHz. We used two different microresonators with similar properties. In both cases, the fibre cavity includes a roughly 1-m polarization-maintaining optical amplifier and a free-space section containing a motorized delay line, polarization control optics to govern the cavity losses and a 12 nm wide bandpass filter.

The optical amplifier's gain medium is a highly doped Erbium fibre (Amonics Ltd). The system now reacts from the 'cold' state in a few seconds, compatible with the bulk nature of our commercial amplifier. The pump power[53–55] changes its non-linear response, a process that has been used in other works to control self-organization in multimode fibres[56,57]. However, it typically only provides a small variation in the total system focusing non-linearity, primarily dominated by a large focusing thermal effect[53–55]. As we discuss further in the Supplementary Information, Section S1 we can exploit this small change because, in our double recirculating cavity design, non-linearities of the same type (focusing in our case, because the thermal non-linearity dominates in both the microcavity and the laser loop) effectively cancel each other, resulting in a relatively small net variation in the non-linearity. In addition, the loss becomes a critical control parameter because it changes the balance of the optical energy between the microresonator and the laser cavity, enabling the system to operate under a different effective non-linearity.

### Dependence of the non-linear gain refractive index on pump power

Erbium amplifiers have a resonance around 1,538 nm and show a strong, step-like non-linear dispersion that changes sign around the resonance and increase in magnitude with pump power until saturation. Specifically, the well-known spectral response of refractive index and gain of Erbium shows a resonant behaviour around 1,538 nm, with the classical, step-like response of the refractive index ruled by Kramers–Kronig relations. In particular, the jump in the refractive index response increases with the magnitude of the gain, resulting in a decrease of the refractive index for wavelengths longer than the resonance and an increase for shorter wavelengths.

Notably, because the gain saturates with the circulating laser power within the fibre cavity, this relationship means that the refractive index dependence with circulating laser power is defocusing for wavelengths shorter than 1,538 nm and focusing for longer. In our experiment, using the intracavity 12 nm filter, we select this portion of the spectral gain. Because the displacement of the refractive index directly depends on the gain, and hence on the pump power, the non-linearity provided by the gain can be controlled with the 980 nm pump. The gain material (and, in general, any gain material) provides then a practical degree of freedom to directly modify the slow non-linear response of the system.

The modelling reported in the Supplementary Information, Section S1 exactly describes this behaviour and is obtained from the very general Maxwell–Bloch relationships that, practically, are the simplest approximation of any gain material. Hence, any gain material can be, in principle, adapted for this purpose.

### Data acquisition

We simultaneously characterized the operating state of our microcomb laser with several instruments, including an optical spectrum analyser (Anritsu), a fast oscilloscope (Lecroy) to retrieve the radio-frequency spectrum, as well as by recording the intracavity power at several locations within the cavity. An autocorrelator (Femtochrome) is used to record the temporal traces and discriminate single-soliton states (with a single pulse within a period of 20 ps) from several soliton states. Typical two-soliton traces are reported in Figs. 1j,l and 4a. Here the autocorrelation shows the typical signature (three peaks) of two identical but not equidistant pulses, as discussed in ref. [21]. Further, we measured the absolute frequency of the oscillating microcomb laser lines using laser scanning spectroscopy in the same configuration as in refs. [21,51] by using a metrological optical frequency comb (Menlo Systems) with the addition of a gas cell for referencing the absolute frequency axis.

The full dataset used to construct the map shown in Fig. 4, with its three repetitions reported in the Supplementary Information, Section S2, is retrieved by an automated procedure of more than roughly 10 h per map, during which all measurements are acquired for roughly 3,500 individual settings within the defined parameter ranges (EDFA pump power and fibre-cavity delay length). We first set the amplifier pump power to zero, then fixed the cavity length, and eventually ramped the amplifier pump power up to the first value. Next, we waited a few seconds for the system to reach the stationary state before obtaining measurements from the instruments previously listed. Next, we increased the pump power in steps of around 1.3 mW, and we repeated the process until reaching the maximum pump power in the range. On completing the set, we turned off the amplifier and repeated the procedure for the next delay stage setting until we probed every point of the parameter space.

For the measurements presented in Fig. 4, we maintained the environment's temperature in the surroundings of the microresonator photonic chip with a proportional–integral–derivative controlled Peltier heater to within ±1° C throughout the experiment. We repeated the data acquisition four times with the same range of parameters, with the repetitions reported in Supplementary Information. It is clear from these repetitions that the soliton regime appears consistently within the same region in the parameter space, yielding the same number of solitons. Across the observed soliton range, the usable output power varies up to 10 mW, with single- and two-soliton states continuously present throughout. For different sets of losses, we obtain two- and three-soliton states with energies reaching up to 30 mW. This range, especially considering the overall optical power, is exceptional and promising for further applications of this laser. It already meets the power requirements of many metrological and telecommunications applications without the need for amplification, which would not be amenable to sustaining broadband pulses.

Finally, we notice that some single-soliton states coexist with a few blue-detuned modes near 1,535 nm. The laser scanning spectroscopy measurements reported in Supplementary Information, Section S2 show that these resonances contain two oscillating lines: one red-detuned (belonging to the soliton) and one blue-detuned (belonging to a superimposed state) for a couple of comb modes in this region. These states seem superimposed over the single comb state and represent the most visible variation in the comb spectra, otherwise unchanged. Hence, these states represent an independent perturbation that does not affect the quality of the soliton state. The spectra and autocorrelations of the two-soliton states indicate that the spacing of these pulses within the microcavity are not generally equidistant. Among our extensive set of experimental data, we have observed a random distribution of the distances of the two-soliton states, often evolving in time. This confirms the localized behaviour of these pulses.

## Characterization of the soliton spectra and numerical fitting

The general properties of a single-soliton state in our system are summarized in the Extended Data Fig. 1, which shows a comprehensive characterization of the different output ports of the microcavity, along with numerical fitting and radio-frequency noise at the given repetition rate.

The experimental data are numerically fit with the mean-field model used in refs. [21,48,51], which consists of a coupled system of dissipative non-linear equations[58–62]. In the Supplementary Information, this model is expanded to add the description of the slow, energy-dependent non-linearities, which explains the findings stemming from the experiments reported in the paper.

A lossy non-linear Schrödinger equation models the evolution of the variable $a$ for the microcavity field in the time and space coordinates $t$ and $x$, normalized, respectively, to the fibre-cavity round-trip and microcavity length. The field in the main amplifying loop $b_0$ corresponds to the leading supermode (that is, the set of modes filtered by the microcavity). A generic supermode is represented by the field $b_q$. The integer $q$ indicates the order of the supermode. The dynamical equations are as follows:

$$\partial_t a = \frac{i\zeta_a}{2}\partial_{xx}a + i\,|a|^2\,a - \kappa a + \sqrt{\kappa}\sum_{q=-N}^{N}b_q, \qquad (1)$$

$$\partial_t b_q = \frac{i\zeta_b}{2}\partial_{xx}b_q + \sigma_6\partial_{6x}b_q + 2\pi i\,(\Delta - q)\,b_q + g\,b_q - \sum_{p=-N}^{N}b_p + \sqrt{\kappa}a. \qquad (2)$$

Here $i$ is the imaginary unit, $q$ and $p$ are supermode indices running from $-N$ to $N$, for a total number of $2N+1$ supermodes, $\partial_{xx}$ and $\partial_{6x}$ are second and sixth-order derivatives. The parameter $\Delta$ represents the normalized frequency detuning between the two cavities, $g$ is the normalized gain and the group-velocity-dispersion coefficients are $\zeta_{a,b}$, with values of $\zeta_a = 1.25 \times 10^{-4}$ and $\zeta_b = 2.5 \times 10^{-4}$. As the gain is tailored with a 12 nm flat-top filter, we use a sixth-order derivative to reproduce the gain dispersion, with $\sigma_6 = (1.5 \times 10^{-4})^3$. The coupling coefficient is $\kappa = 1.5\pi$. Further details are reported in Supplementary Information, Section S1. These parameters are used to fit the experimental spectra in the Extended Data Fig. 1 with a numerical mode solver that provides the non-linear eigenmodes of the system, including the soliton functions, as in refs. [21,48,51]. In particular, the field measured at the 'drop' port directly reports the microcavity internal field $a$. The field at the output port, which is the 'through' port of the microcavity, can be theoretically evaluated with $c(t) \approx b_0 - \sqrt{\kappa}a$. To fit the experimental data, we included an extra component $\alpha b_0$ to this value, where $\alpha$ is a coefficient that accounts for the non-ideal transmission of the microcavity and the polarization interference at resonance. The numerical fit of a typical experimental spectrum of this output is reported in Extended Data Fig. 1a,b,d,e for the soliton spectra in Figs. 1c and 2f. Here we also present the experimental measurement of the gain + 12 nm filter bandwidth, showing how the soliton spectrum well exceeds the amplification spectrum.

For the states in Extended Data Fig. 1a,b,d,e, the input powers to the microcavity were 44 and 63 mW, respectively. The 'through' output powers were instead 4 and 5 mW. The second output (drop port) was reconnected to the amplifier leading to off-chip emitted powers of 6.5 and 8.3 mW, respectively. Part of the light was extracted for characterization with a beam splitter. The total cavity operated with roughly 10 dB gain. When accounting for the on-chip losses of 3 dB, we estimated that the microresonator operated with 31 and 44 mW on-chip input powers. The on-chip 'through' output powers were 5.7 and 7.1 mW, whereas the on-chip 'drop' port powers were 9.3 and 11.8 mW. This results in on-chip non-linear conversion efficiencies of about 20 and 30% at the 'through' and 'drop' ports, respectively.

Finally, Extended Data Fig. 1g,h reports the radio-frequency characterization around the repetition rate frequency. A small portion of the output 'through' port signal was processed with an electro-optic modulator, leading to further sidebands around each of the original comb lines. As the electro-optic modulator was driven in saturation with a GPS-referenced microwave oscillator, several harmonic sidebands were generated, whose frequency distance from the comb lines was a multiple of the modulating signal frequency[63]. We considered the third harmonic sidebands, and we set the modulation frequency such that the interaction between adjacent comb lines produced a $f_0 = 500$ MHz beat note. We showed it with an amplified photo-detector and analysed it with an electrical spectrum analyser (ESA). Evidently, in all the ESA traces, the repetition rate beat-note signal to noise ratio is more than 40 dB, here being limited mainly by the ESA noise floor.

## Data availability

Data generated and analysed in the main text figures are available at Figshare, https://doi.org/10.6084/m9.figshare.19755313. Additional datasets generated during the current study are available from the corresponding author on reasonable request.

## Code availability

The codes used during the current study are available from the corresponding author on reasonable request.

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

**Acknowledgements** We acknowledge the support of the EPSRC, Industrial Innovation Fellowship Programme, under grant no. EP/S001018/1, the UK Canada Quantum Technology Programme and Innovate UK (IUK project nos. 77087 and 10004412). This project has received funding from the European Research Council (ERC) under the European Union's Horizon 2020 research and innovation programme grant agreement no. 851758 (TELSCOMBE). A.C. acknowledges the support of the DSTL-Defence Science & Technology Laboratory through the studentship DSTLX1000142078. J.S.T.G. acknowledges the Leverhulme Trust (Leverhulme Early Career Fellowship grant no. ECF-2020-537). R.M. acknowledges funding by the Natural Sciences and Engineering Research Council of Canada (NSERC) through the joint UK Canada Quantum Technology Programme, and by the Canada Research Chair Program. B.E.L. acknowledges support from the Strategic Priority Research Programme of the Chinese Academy of Sciences (grant no. XDB24030300). We are indebted to L. Peters, L. Olivieri and A. Bendahmane for enlightening discussions.

**Author contributions** M.R. and A.P. developed the original research idea. B.E.L. and S.T.C. designed and fabricated the integrated devices. M.R., P-H.H. and H.B. designed the experimental setup. M.R., P-H.H. and A.C. performed the experiments. A.P., A.C., J.S.T.G. and G-L.O. developed the theoretical model and ran the numerical analysis. M.R., A.C., P-H.H. and A.P. analysed the experimental data. All the authors contributed to the development of the experiment, numerical model and to data analysis. M.R., J.S.T.G., D.J.M., R.M. and A.P. drafted the main paper, and all the authors contributed to the writing of the manuscript. A.P. and M.P. supervised the research.

**Competing interests** The authors declare no competing interests.

**Additional information**
**Correspondence and requests for materials** should be addressed to Alessia Pasquazi.

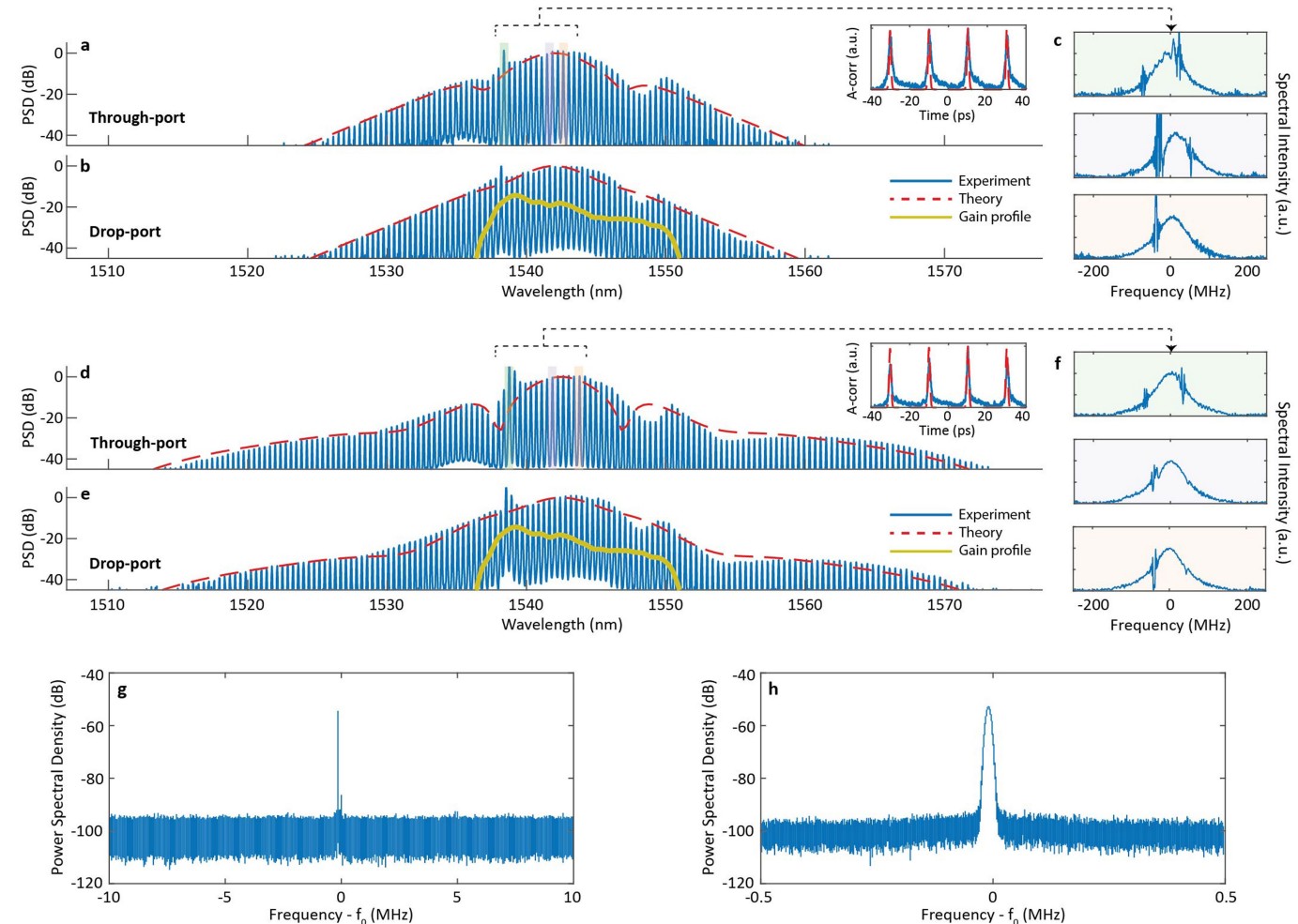

**Extended Data Fig. 1 | Characterization of the soliton state. a–c**, Example soliton state, obtained with a 350 mW pump at 980 nm. Experimentally obtained optical spectra of a single soliton (blue) with their theoretical fit (red dashed) at the 'through' (panel **a**) and 'drop' (panel **b**) ports. In the fit, we use $g = 0.08$ and $\Delta = 0.46$, with the other parameters reported in the Methods. The operational EDFA gain bandwidth is indicated by the overlay in yellow. The input power to the microchip is 44 mW and the measured output powers are 4 mW at the 'through', and 6.5 mW in the intracavity 'drop' ports, respectively. The insets correspond to the autocorrelation trace, clearly showing one peak per microcavity round-trip. **c**, Three examples of the intracavity spectrum (blue), displaying the lasing modes within each microcavity resonance and corresponding to the highlighted wavelengths. While the modes are all red detuned, we observe in the mode around 1538 nm, where the peak of the Erbium is located, the coexistence of the soliton red detuned mode with a blue-detuned continuous wave mode in panel **a**. **d-f**, Same

as panels **a–c**, for a single soliton with a slightly higher pump power of 360 mW at 980 nm. In the fit we use $g = 0.1$ and $\Delta = 0.4$, with the other parameter reported in the methods. The input power to the microchip is 63 mW and the measured output powers are 5 mW at the 'through', and 8.3 mW at the intracavity 'drop' ports. The spectral shape is theoretically fitted by the same model. **e**, Same as panel **c**, for the spectrum shown in panel **d**. **g**, Radio-frequency spectrum measured around the repetition rate frequency via heterodyne modulation. A small portion of the 'through' output signal is processed with an electro-optic modulator, leading to additional sidebands around each of the original comb lines. We considered the third harmonic sidebands, and then set the modulation frequency such that the interaction between adjacent comb-lines produced a $f_0 = 500$ MHz beat-note. Electrical spectrum analyser trace of the signal derived from the amplified photodiode after the electroptic modulation, at 50 MHz span, and resolution bandwidth of 30 kHz. **h**, Same as panel **g** but with 1 MHz span, and resolution bandwidth of 10 kHz.

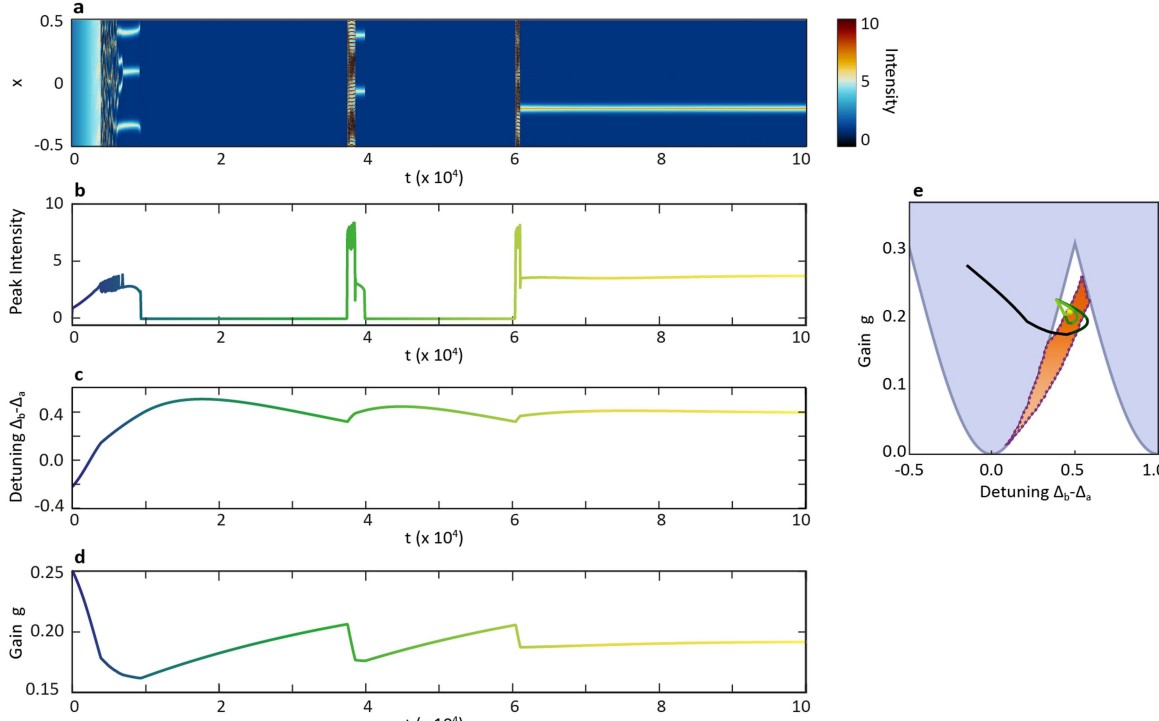

**Extended Data Fig. 2 | Start-up in the presence of gain-induced nonlinearity and thermal nonlinearity: single soliton formation.** Numerical propagation for Eqs. (1–4) in Supplementary Section S1, modelling a microresonator-filtered fibre laser with the inclusion of a saturable gain and gain-induced nonlinearity in the amplifying cavity, and a thermal nonlinearity in the microcavity. The system parameters are $\Gamma_T = 5$, $\tau_T = 8 \times 10^3$, $\eta = 0.4$, $\tau_g = 4 \times 10^4$, $\Theta = -13$, $g = 0.25$ and $\Delta + \Theta g_P = -0.21$. **a**, Pseudo-colour map of the electric field intensity in the microcavity, in the normalized units of Eqs. (1–4), as a function of the position in the microresonator $x$ (which is normalized against the microcavity roundtrips) and time $t$ (which is normalized against the main-cavity roundtrips). **b**, Temporal evolution of the peak intensity. The colours varying for increasing times matches with the plot inside panel **e** (showing the attractor). **c**, Temporal evolution of the effective detuning $\Delta_b$-$\Delta_a$. The colours varying for increasing times matches with the plot in panel **e**. **d**, Temporal evolution of the gain. The colours varying for increasing times matches with the plot in panel **e**. **e**, Map of the attractor for the peak intensity, detuning and gain, as in **c**, d, following the colour code of **b**–**d** as a function of time. The attractor is superimposed to the soliton (orange) and zero state (blue) stability regions. Figure 2a of the main text reproduces this map.

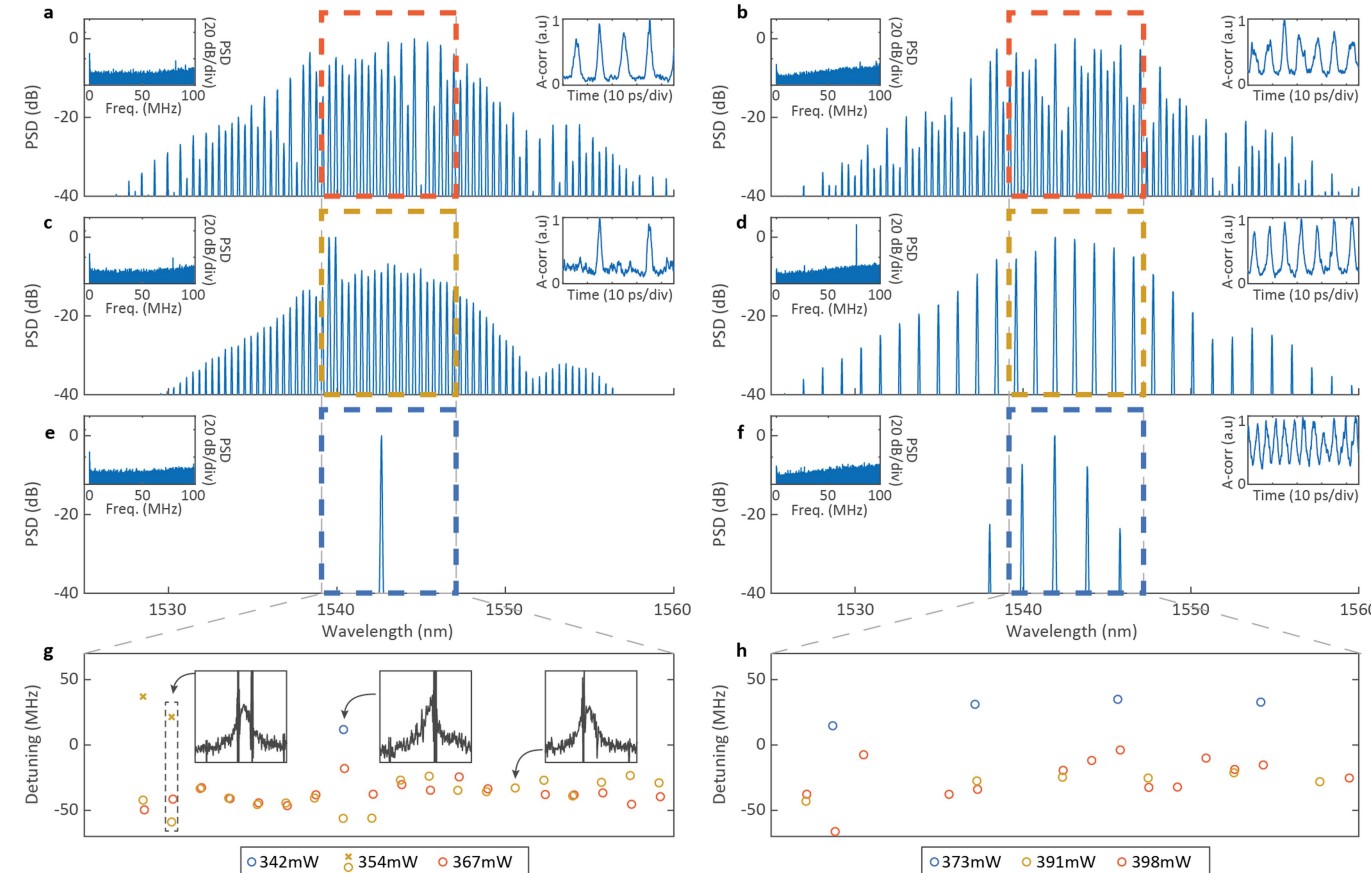

**Extended Data Fig. 3 | Summary of laser scanning-spectroscopy measurements. a**, Power spectral density (PSD) of optical and radio-frequency spectra (left inset) with autocorrelation (right inset) for an EDFA pump power of 367 mW, for the system with 16 dB losses. **b**, Same as panel a, for an EDFA pump power of 398 mW, for the system with 14 dB losses. **c**, Same as panel **a**, for an EDFA pump power of 354 mW. **d**, Same as panel **b** for an EDFA pump power of 391 mW. **e**, Same as panel **a**, for an EDFA power of 342 mW. **f**, Same as panel **b** for an EDFA power of 373 mW. **g**, The insets show three typical laser scanning spectroscopy measurements for a red-detuned oscillating line (right), blue-detuned line (centre), and coexistence of two oscillating modes (left).

From these measurements, it is possible to extract the absolute frequency position of the oscillating line and the centre of the microcavity resonance. We define their difference as the frequency detuning of the lasing states in operating conditions. The distribution of the individual mode detunings across the wavelength span highlighted by the grey dashed lines for 16 dB intracavity losses and EDFA pump powers of 342 mW, 354 mW and 367 mW are shown in blue, yellow, and red, respectively. **h**, Same as panel **g**, for the system at 14 dB intracavity losses, and for EDFA pump powers of 373 mW, 391 mW and 398 mW.

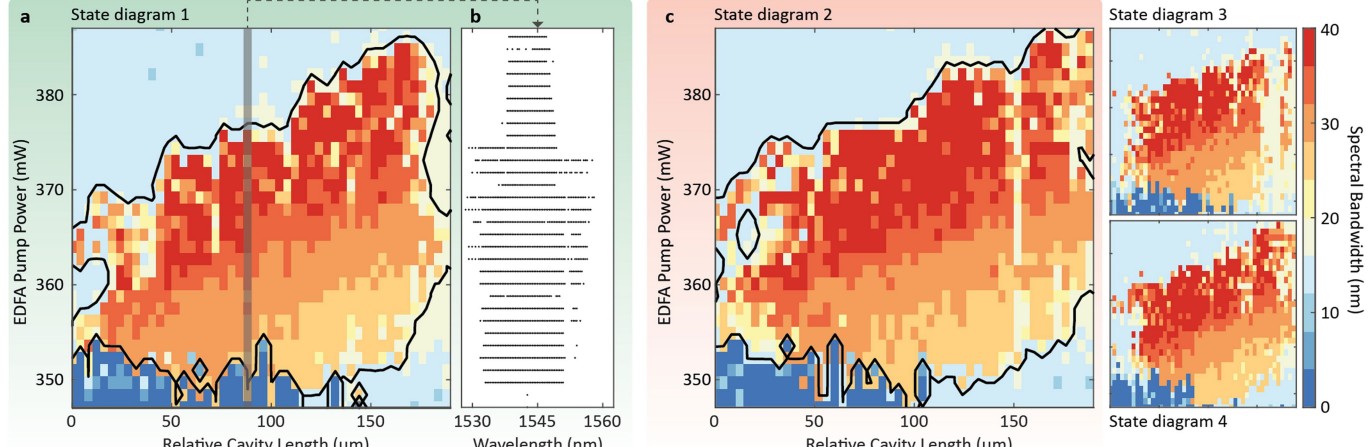

**Extended Data Fig. 4 | Repetitions of experimental state diagrams for the microresonator-filtered fibre laser with intracavity losses set to 16 dB.** **a**, Optical spectral bandwidth (calculated as the bandwidth at -40 dB from the maximum) of the laser states, as a function of the cavity length and EDFA pump power. The soliton states are found in the yellow to red region with the broadest bandwidths, while CW states are in blue. Outside these regions, roughly defined by the black lines, the lasing regime is unstable. During the 10-h

experiment performed to acquire the data, the temperature was maintained at 40°C to within a fluctuation of a few degrees. **b**, Optical spectra of the states along with the 92 μm delay position, identified by the grey line in panel **a**. The black dots represent the frequency position relative to the maximum value of each comb mode. **c**, Repetition of the same experimental map, with two further repetitions on the insets, acquired immediately after the map in panel **a**, over subsequent 10-h long experiments.

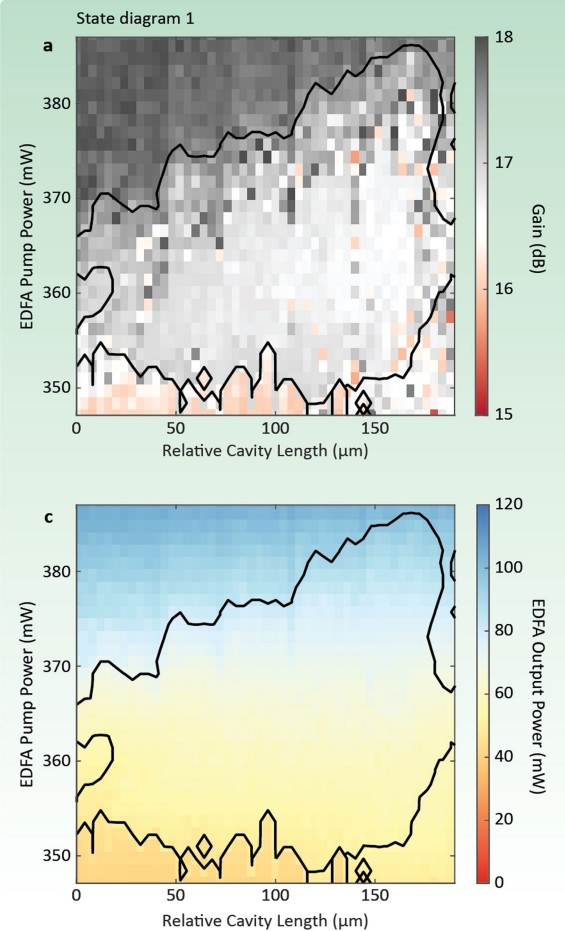

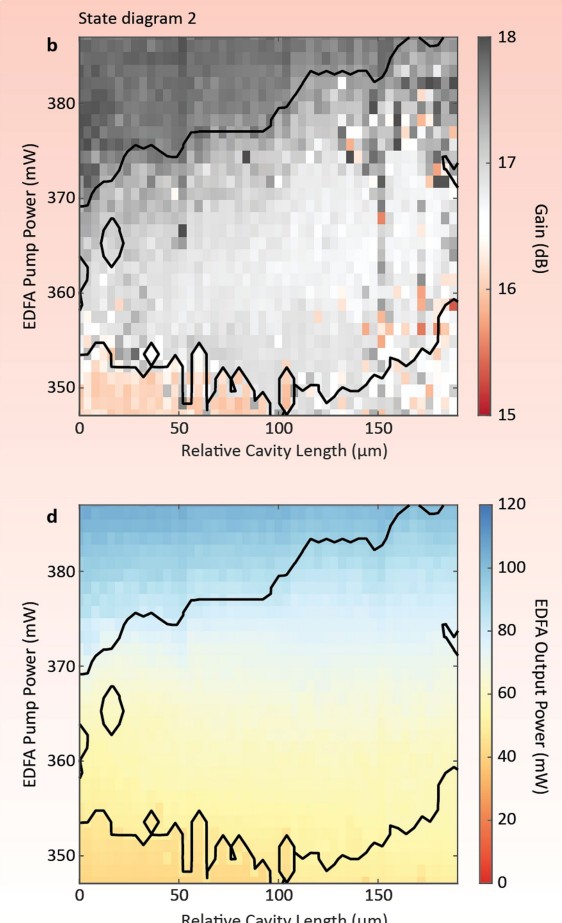

**Extended Data Fig. 5 | State diagram optical gain and intracavity power measurements.** Additional measurements for the 'State Diagram 1 and 2' presented in Fig. S4a and c, respectively. **a**, Optical gain of the EDFA for the 'State Diagram 1'. **b**, Same as panel **a**, for the 'State Diagram 2'. **c**, Intracavity power measured at the output of the amplifier or, equivalently, at the input of the microcavity, for the 'State Diagram 1'. **d**, Same as panel **c**, for the 'State Diagram 2'.

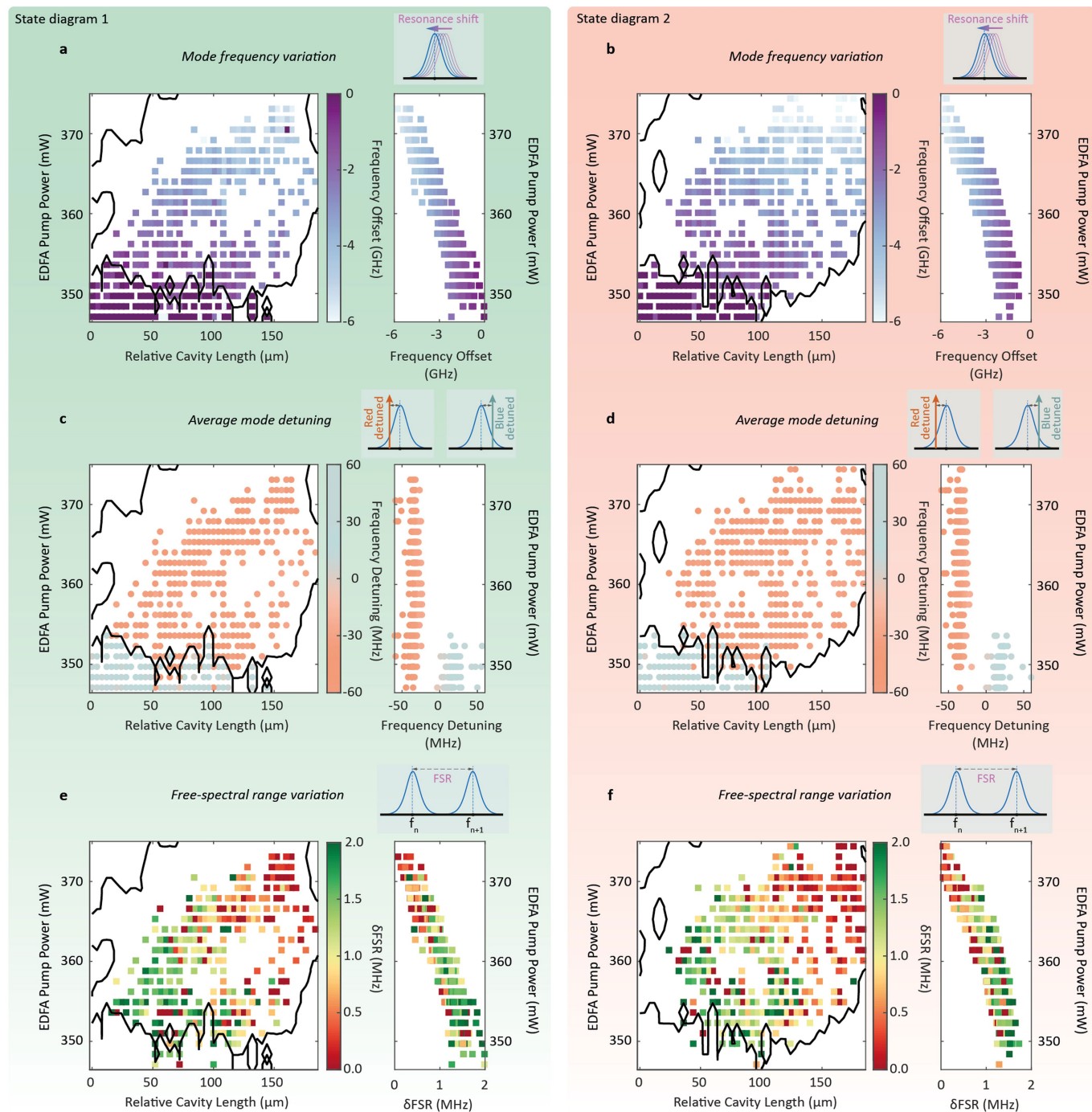

**Extended Data Fig. 6 | Analysis of the frequency positions of the stable states.** Data extracted from the laser scanning spectroscopy. Black lines mark soliton and CW region boundaries. **a**, Variation of the frequency position of a selected resonance of the microcavity (here, we choose the mode centred approximately at 1,543 nm). The colour code in the map follows the inset (right), showing the values of the frequency shift for the microcavity resonance. **b**, Same as panel a, now for the 'State Diagram 2'. **c**, Variation of the average detunings, calculated as the mean frequency difference between each oscillating microcomb laser line and the corresponding microcavity resonance centre across all microcomb lines. The colour code in the map follows the colours of the inset (right), where the values of the detunings versus EDFA pump power are shown. **d**, Same as panel c, but for the 'State Diagram 2'. **e**, Free-spectral range (FSR) variation associated with the soliton states (FSR~48.9 GHz) obtained across the mapped values. The colour code in the map follows the inset (right), showing the free-spectral range variation values. Same as panel **e**, but for the 'State Diagram 2'.

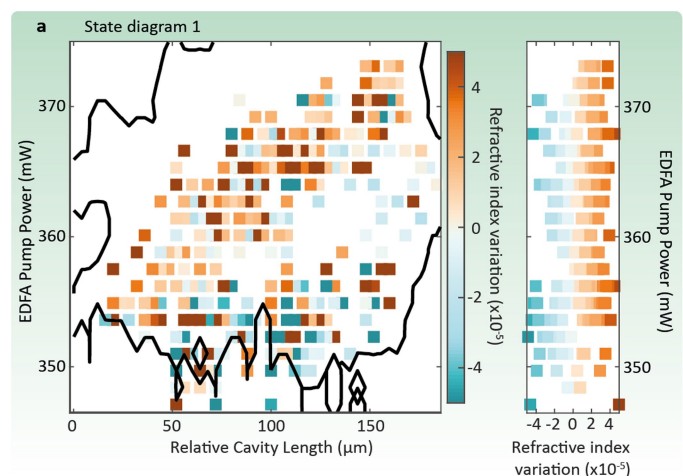

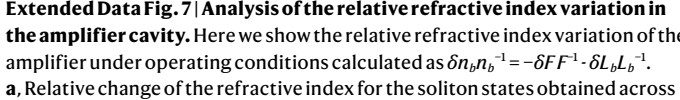

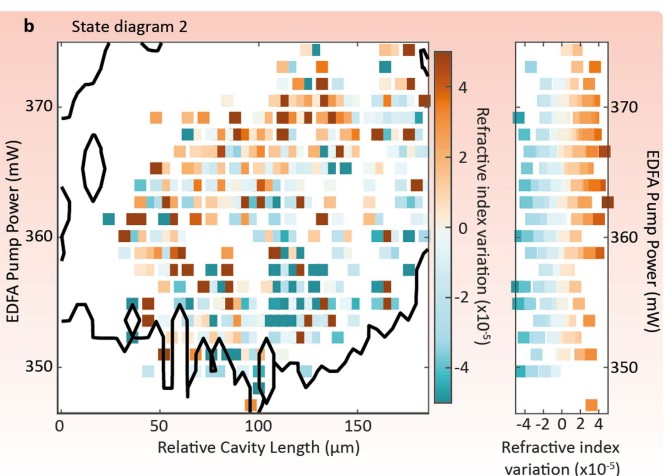

**Extended Data Fig. 7 | Analysis of the relative refractive index variation in the amplifier cavity.** Here we show the relative refractive index variation of the amplifier under operating conditions calculated as $\delta n_b n_b^{-1} = -\delta F F^{-1} \cdot \delta L_b L_b^{-1}$. **a**, Relative change of the refractive index for the soliton states obtained across the mapped values, extracted from the free-spectral range variation of the 'State Diagram 1'. The colour code in the map follows the inset showing the refractive index change. **b**, Same as panel **a** for the 'State Diagram 2'.

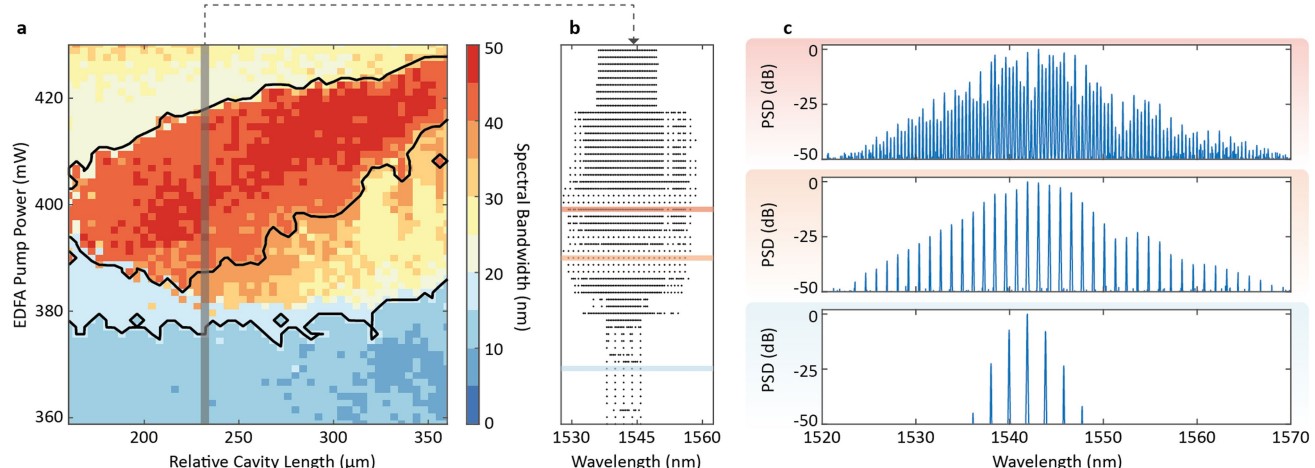

**Extended Data Fig. 8 | Experimental state diagram for a microresonator-filtered fibre laser with intracavity losses set to 14 dB.** The delay axis uses the same zero definition as Fig. S4. **a**, Spectral bandwidth (calculated as the bandwidth at -40dB from the maximum), in colour code, of the laser state as a function of the cavity length delay and EDFA pump power. Soliton states are found in the red region with the broadest bandwidths, while Turing patterns appear at the lowest EDFA values (blue). The lasing regime is unstable outside these regions, which are roughly defined by the black border lines. During the experimental acquisition, the temperature was maintained at 40 °C within a fluctuation of a few degrees. **b**, Optical spectra of the states at the delay 240 μm, identified by the vertical grey line in panel **a**. The black dots represent the frequency position relative to the maximum value of each comb mode. **c**, From top to bottom, power spectral densities (PSDs) of the optical spectra for EDFA powers of 398 mW, 391 mW and 373 mW, respectively, as indicated by the corresponding coloured highlights in panel **b**.

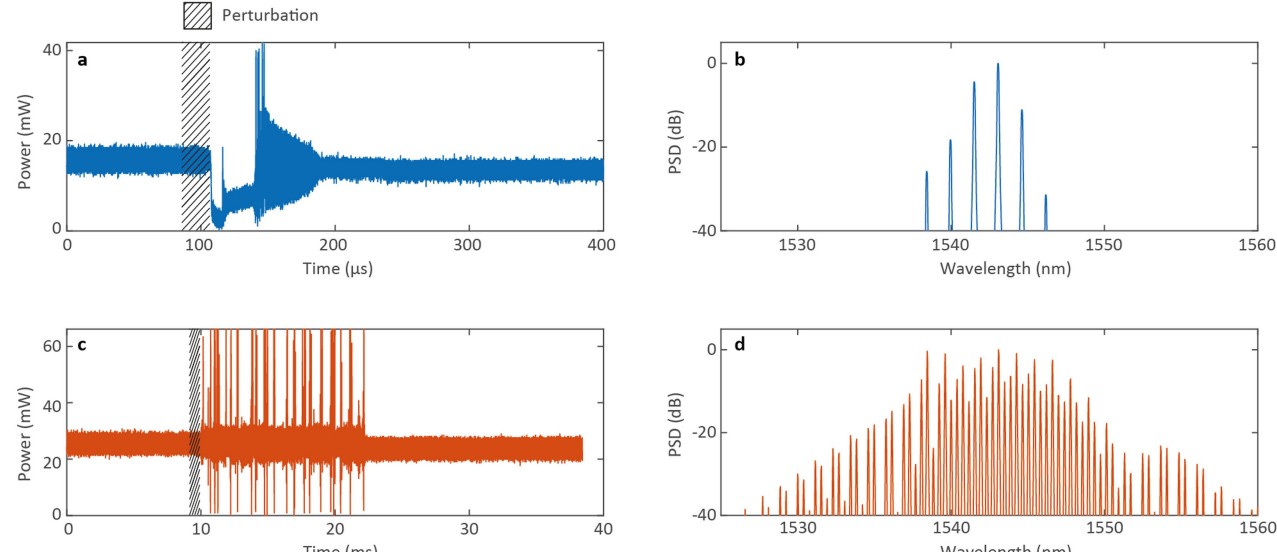

**Extended Data Fig. 9 | Perturbation recovery.** The stationary state is perturbed with a steep variation in the voltage driving the EDFA pump power. **a**, Recovery of a Turing pattern state. The power associated with the state is shown in blue, while the black shaded region indicates the time at which the EDFA pump power is modified to perturb the state. The system recovers to a state with similar average power and spectral shape after 0.1 ms, which is in the temporal scale of the microcavity thermal nonlinearity response. **b**, Corresponding optical spectrum of the recovered state in **a**. **c**, Same as panel **a**, for a multi-soliton state. Note that the recovery time is on the order of 10 ms, dictated by the Erbium gain. **d**, Corresponding optical spectrum for the recovered state in panel **c**.

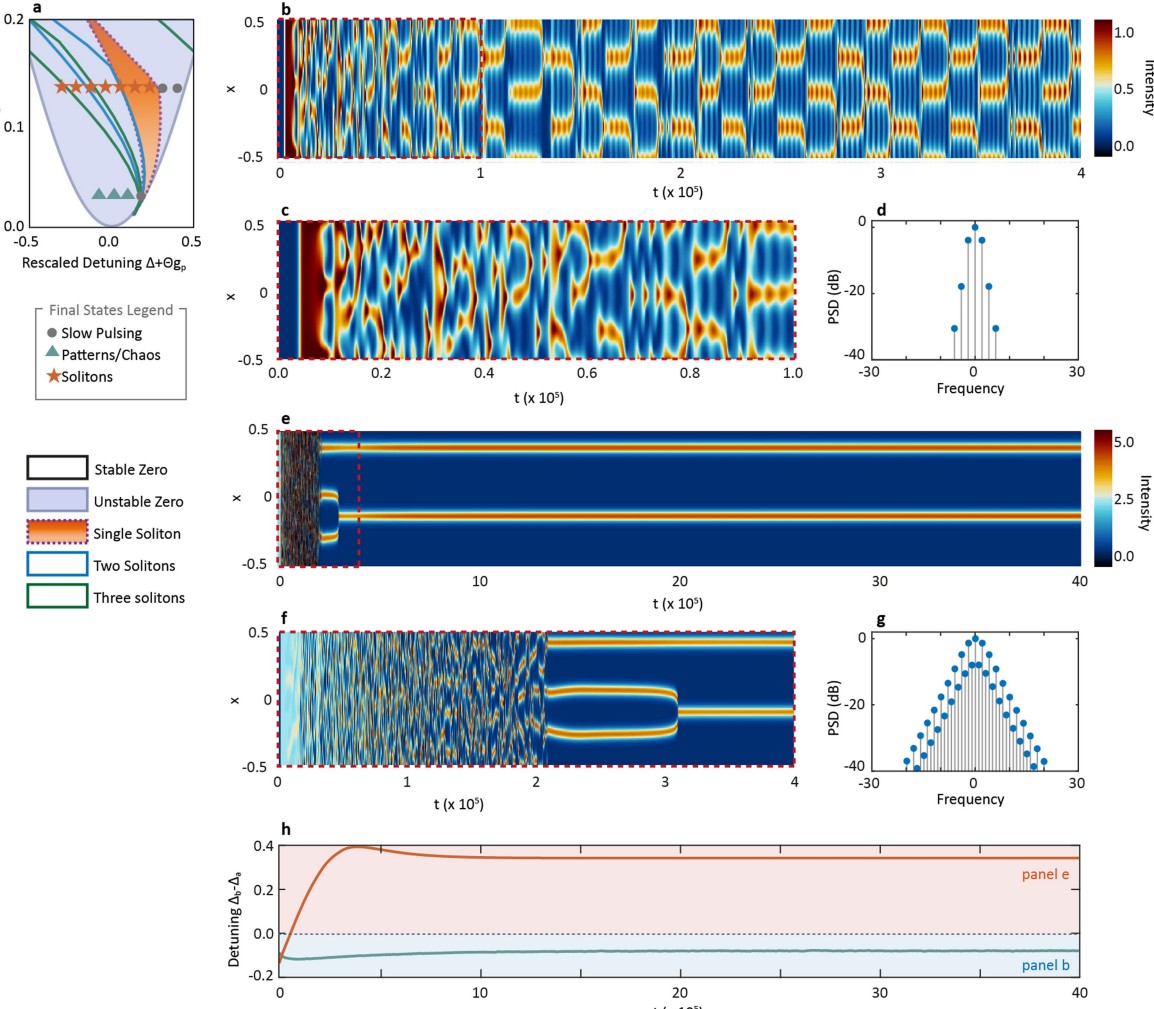

**Extended Data Fig. 10 | Start-up in the presence of gain-induced nonlinearity and thermal nonlinearity. a**, Stability region and a selection of steady states for the system. The colour of each marker represents the type of steady-state solution reached by the system after start-up from the noise for different values of the rescaled detuning $\Delta + \Theta g_P$ and gain $g$. **b**, Pseudo-colour map of the electric field intensity in the microcavity as a function of the position in the microresonator $x$ (normalized against the microcavity roundtrips) and time $t$ (normalized against main-cavity roundtrips). The system parameters are $\Gamma_T = 2$, $\tau_T = 8 \times 10^3$, $\eta = 0.1$, $\tau_g = 4 \times 10^4$, $\Theta = -15$, $g = 0.03$ and $\Delta + \Theta g_P = -0.0909$. **c**, Enlarged view of panel **b** for the first $10^4$ main-cavity roundtrips, highlighting the start-up of the pattern-like solution within a timescale compatible with the thermal time constant $\tau_T = 8 \times 10^3$. **d**, Output power spectral density (PSD) of the $a$ field. **e**–**g** Same as panels **b,c**, **d**, but for parameters $\Gamma_T = 2$, $\tau_T = 8 \times 10^3$, $\eta = 0.1$, $\tau_g = 4 \times 10^4$, $\Theta = -15$, $g = 0.135$ and $\Delta + \Theta g_P = -0.1342$. **f**, Illustrates the start-up of the two-soliton state from noise within a timeframe compatible with the slower timescale $\tau_g = 4 \times 10^4$. **h**, Temporal evolution of the effective detuning $\Delta_b$-$\Delta_a$ for the low-gain pattern-like states (blue line) and the two-soliton states (orange line).