## [Peer Review File · Nature]

Manuscript Title: Self-emergence of robust solitons in a micro-cavity.

Reviewer Comments & Author Rebuttals

Reviewer Reports on the Initial Version:

Referees' comments:

Referee #1 (Remarks to the Author):

The authors demonstrate an approach for spontaneous generation of solitons in a micro-resonator-filtered fiber laser system, taking advantage of the gain of the erbium-doped fiber and the thermal nonlinearity to create an attractor. By controlling the fiber cavity delay and the pump power, they find the region over which a stable soliton state can be generated. Using this approach, they generate a soliton state spanning a bandwidth of 30 nm. The major outstanding challenge in the community is not to repeatedly get into some arbitrary soliton state, rather it is to get into the specific soliton state each time, particularly a single-soliton state. The results indicated in this manuscript do not reflect this and show that it reaches a different state depending on the EDFA power implying that an initial parameter sweep is necessary for generating the specific soliton state which is no different from other approaches. The authors also state that the number of solitons is controlled by the loss of the cavity and accessible through control of the intracavity energy. This laser-cavity soliton approach has been intensely studied by the authors and published elsewhere (e.g. Bao, et al., Laser cavity-soliton microcombs, Nat. Photonics 13, 284 (2019).) and theoretical analysis has also been performed in Cutrona, et al., Temporal cavity solitons in a laser-based microcomb: a path to a self-starting pulsed laser without saturable absorption, Opt. Express 29, 6629 (2021). Thus, this work is too incremental and I cannot recommend this paper for publication in Nature. Specific comments are below.

1. The authors suggest that their approach is more robust as compared to the previously demonstrated approaches but the previous approaches consistently can achieve the same single soliton state based on their parameter settings. The present manuscript does not successfully achieve the same state implying that some configuration is also necessary to reach a certain single soliton or multi-soliton state. Can the authors clarify this point?
2. It seems that, for a range of power and delay there's an existence range for the soliton state. How much do the microresonator parameters affect the range? The generated combs are quite narrowband, does the existence range change for broader comb spectra? Can a broader comb be generated? How universal is the approach?
3. In Fig. 2C, middle plot, it seems that that this state may be a single-soliton state? Can the authors comment on the modulation in the spectrum that has a minimum near 1550 nm?
4. In Fig. 5, the authors show the stability of the system by perturbing the pump power. How much of a perturbation is allowed for the system to return to the original Turing or soliton state?

Referee #2 (Remarks to the Author):

Rowley et al. present a thorough experimental study, backed by numerical simulations, showing that a micro-cavity embedded into a simple fibre ring laser can lead to a self-starting cavity soliton operation.

The cavity soliton terminology traditionally refers to a solitary pulse on background that is solution

of the Lugiato-Lefever equation, which is not precisely the case in this work, as the generated solitons have more attributes of bright laser solitons.

As a matter of fact, the authors emphasize on the advantage of these pulses with vanishing tails that allow a superior energy conversion ratio from the pump to the multiple comb lines, and I fully agree with them in that respect. Nevertheless, keeping the cavity soliton terminology for laser pulses brings some confusion. From the laser point of view, we rather have a situation of high-harmonic mode locking that is promoted by filter-driven four-wave mixing from an embedded micro-cavity.

Otherwise, the fact that such laser cavity can achieve a self-starting harmonic mode locking without the assistance of an obvious saturable absorber mechanism is truly impressive. The physical mechanisms involve a delicate combination of two slow (thermal and gain) nonlinearities to move the cavities resonances during the laser start-up phase and enable a soliton operation involving the fast Kerr nonlinearity and the chromatic dispersion.

The authors have an impressive expertise in such physical system and its modelling, well manifested for instance in their 2019 Nature Photonics article on "Laser cavity-soliton microcombs". However, I am not convinced that the present results would be suitable to Nature. Laser source performances are comparable to these of the 2019 paper. After a very good introduction on the fundamental challenges met by the control of cavity-soliton sources, the highlighted novelty is summarized in simple terms. Then, the following of the manuscript becomes highly technical and not clearly written, requiring the reader to go back and forth with efforts from the figures to the text and the supplementary materials. Among several possible examples, that of Figure 1c representing a numerical simulation which is not discussed, comes probably too early, and requires going through the supplementary material section. Basically, the manuscript is not entirely self-consistent and the interpretations of the figures are too scarce, while heaps of data are displayed on experimental maps. However, important experimental data is missing, such as detailed characterization of the laser output in the time and radio-frequency domain for a selection of regimes (soliton state, Turing pattern...).

The supplementary material section provides numerous information but looks like a series of internal reports lacking fluidity. I also noticed dubious parameter values, such as $\beta_b \approx -60 \text{ ps}^2 \text{ km}^{-1}$ for the optical fibre and band-pass filter bandwidth $\Delta F_F = 650 \text{ THz}$, but I could be wrong of course.

To conclude, I find the main text, past the introduction, needs a significant polishing. I find that the present manuscript is not, in its present form, suitable to the general readership of Nature but should rather be submitted to a Photonics journal.

Referee #3 (Remarks to the Author):

The paper reports the development of a robust source of cavity solitons where a micro-resonator is used as a filter within a fibre laser cavity. If I understand correctly, the nonlinearities in the two systems (micro-resonator and laser cavity) act to balance each other, allowing for the stable formation of cavity solitons over a reasonably large parameter space. This, to me, seems to be a significant technological advance in terms of the development of systems for use in real-world applications. The results look convincing (well, at least those that I can understand) and I am not in any doubt that the system does what they claim. Where I am struggling a little is understanding some of the details of the results as I find the presentation quite technical for a general audience. In this regard, I have provided a series of comments that I feel would be helpful if the authors could address during their revisions.

1) The figures are very technical and the paper could really do with some schematics that help to illustrate the key points that would be understandable to a broad readership.

-For example, I cannot work out what Fig. 1c is trying to show – why is the plot inverted (red peaks are actually zero, blue background is maximum?) and what sort of round trip numbers are represented in the x and y axes (1 or many?). Does this even matter? I think the authors need to think clearly about what information they are trying to present here and whether it is accessible to readers without having to work too hard.

-Also, if I understand correctly, Fig 3 is supposed to show the flexibility of the system. However, this was lost on me the first two times I looked at this image. I wonder if there is a better way to highlight this? Maybe the insets could be brought out of the subfigures and made clearer? Or actually show some real data corresponding to points in the graph?

2) What are the key performance metrics for the soliton sources and what sort of applications could these find use in as the system is currently built? I.e., that are the power levels and/or pulse durations, and are these useful? And where is the output coupler? Is this in the fibre loop?

3) Another key point that the authors stress is the robustness of the system. In this regard it would be helpful to know how robust the operation is to environmental disturbances and temperature fluctuations. In particular, does the dependency on the thermal nonlinearity in the micro-resonator place any limitations on the operation of the system? How long can the system remain in stable, and consistent, operation? Hours, days, longer?

4) The system is sold as a robust, turnkey, source of cavity-solitons for practical applications, but how practical is the set up? The schematic in Fig 1a looks fairly compact and robust, but are the fibres really integrated with the chip? And what about the free-space section? How does it compare with the system published in Ref. [Nature 582, 365–369 (2020)] for size and practicality?

5) I normally expect to be able to find technical details (fibre lengths, resonator size, pump wavelength and power levels etc.) somewhere in the paper or supplementary information. Maybe I missed it, but this information was not obvious to me.

Author Rebuttals to Initial Comments:

We thank the Referees for their comments, which we took extremely seriously. They pointed out a number of issues with the paper that needed to be solved and we believe these comments have really helped us to improve the paper.

In particular, while we had reported single solitons, we acknowledge that this was not highlighted well enough, being largely discussed only in the supplementary. We agree that this is indeed a key issue. Also, the spontaneous start-up from noise was not explicitly demonstrated for the single soliton state. We have now addressed all of these issues, helped by the addition of new significant results stemming from the experiments that we performed in response to the Referee's reports.

We are also grateful to the Referees for frankly stating that our presentation was too technical and could have been more accessible. In response to the Referee's comments:

1. We devised and performed extensive new experiments.
2. These new results now clearly and conclusively show that we achieve consistent and reliable spontaneous start-up into soliton states.
3. We demonstrate that we can startup the system repeatably and controllably into a desired soliton state, including, in particular, single solitons.
4. Our new experiments clearly demonstrate the robust and spontaneous re-formation or recovery of soliton operation states in response to extreme perturbations that fully erase the soliton operation. Moreover, the system always recovers exactly to the same state that it was operating in, naturally and rapidly, without any external control.
5. We have redone the figures and revised the text to reflect the new data and better convey both the main advances of our work and the operation principles of our system more clearly. In addition, we have revised the title to "*Natural Emergence of Robust Micro Cavity-Solitons*".

We respond to all the individual comments below in detail. First, however, we believe it is helpful to summarise the major breakthroughs of our paper, clearly detailed in the answers to the Referee's comments. In summary:

1. We demonstrate the natural, robust, and repeatable formation of cavity solitons, a phenomenon that is essentially independent of pump dynamics, initial conditions, or phases. Our system simply needs to be turned on, and solitons are automatically generated.
2. We achieve this natural emergence repeatably, robustly, and controllably into any of the soliton states that we observe in our system. These include single solitons, determined simply by judiciously setting the pump power. This is a fixed and constant parameter that provides a critical degree of freedom, enabling the selection of the desired state. It is constant for the device unless the desired state of operation changes. It only needs to be set once – “set-and-forget”.
3. We achieve intrinsic, natural and robust stability and immunity to perturbations. In fact, more than just being stable and resilient to perturbations, our solitons actually fully and spontaneously reform to the state they were operating in, even after being completely disrupted, or erased. Our new experimental results (Fig.3) clearly illustrate this. We also demonstrate continuous operation over long timescales.
4. We introduce entirely new and innovative fundamental physics, adding a new dimension to the field of microcombs – a dominant nonlinear attractor – a hallmark of complex, chaotic and nonlinear systems. This actually exploits the thermal nonlinearities of the system to our advantage, enabling us to dramatically scale the power and energy.
5. In other systems, thermal effects pose fundamental limitations. We provide an approach to harness the thermal nonlinearity to the microcomb advantage. Because of this, our method is intrinsically scalable in energy and power. In perspective, this paves the way to microcomb sources that, intrinsically, do not need further amplification. Already in this paper, we generate substantially higher power with much higher efficiency than in other systems. These are critically outstanding performance issues for microcombs.

All of these achievements are significant and unprecedented. We summarise below the point by point answer to each of the Referee's points.

Referee #1 (Remarks to the Author):

The authors demonstrate an approach for spontaneous generation of solitons in a micro-resonator-filtered fiber laser system, taking advantage of the gain of the erbium-doped fiber and the thermal nonlinearity to create an attractor. By controlling the fiber cavity delay and the pump power, they find the region over which a stable soliton state can be generated. Using this approach, they generate a soliton state spanning a bandwidth of 30 nm. The major outstanding challenge in the community is not to repeatedly get into some arbitrary soliton state, rather it is to get into the specific soliton state each time, particularly a single-soliton state.

We thank the referee for the constructive review.

Of course, we agree that *'to get into the specific soliton state each time, particularly a single-soliton state'* is the most important challenge in the literature. As mentioned above, we do acknowledge that this issue could have been dealt with greater detail in our original paper and have fully addressed this now.

While in our original submission we had reported single solitons, this was not highlighted well enough and discussed only in the Supplementary Material. Also, the spontaneous start-up from noise was not demonstrated for the single soliton state.

We have performed new experiments and revised the presentation of our original data to highlight this. We now demonstrate the spontaneous formation (from noise) of all states – both single and multiple solitons – conclusively. Indeed, this is the central result of our paper.

Most importantly, our system start-up is largely independent of the pump dynamics – the system merely needs to be turned on. The state of operation is simply and easily selected by a one-time “set-and-forget” fixed parameter – the pump power. We believe that this achievement is transformative compared to previous efforts.

In addition to all of this, we demonstrate that our system – operating in a single soliton state - is intrinsically robust to perturbations or even total disruption – fully recovering to the same state of operation on its own.

These issues, together with the power output and energy efficiency that we also address, represent the greatest challenges facing microcombs, and we have demonstrated all of them simultaneously and for the first time.

The Referee makes some very valid points, and we are grateful for their input, which has motivated us to fill in some of the gaps in the experimental results and to revise the writing, figures and presentation of the paper to make our results much clearer to the reader.

We now clearly remark in the Introduction:

'A robust, repeatable approach for initiating and reliably maintaining the microcomb into the same type of soliton state, particularly the single soliton state, is widely acknowledged as being critical, with notable progress reported⁸⁻²⁰. Nonetheless, it largely remains the major outstanding challenge confronting this field.'

As we state above, this is exactly what we demonstrate in this work.

We have taken the Referee's remarks very seriously. We have largely revised the paper adding new experimental results that unequivocally show a consistent start-up (Fig. 1 and 2), recovery after perturbation and robustness of the same single-soliton state (Fig. 3). Please see our extensive revisions, which we will describe in our detailed answers below.

The results indicated in this manuscript do not reflect this and show that it reaches a different state depending on the EDFA power implying that an initial parameter sweep is necessary for generating the specific soliton state which is no different from other approaches.

We acknowledge that our previous presentation was not sufficiently clear, and we have provided a better experimental clarification of this point.

As the Referee will now see regarding our new experimental evidence, our method is deeply transformative compared to other approaches. Remarkably, the system behaviour does not depend on the start-up pump dynamics – i.e., a “sweeping” of the pump power or any other parameter. On the contrary, the system merely needs to be turned on. Further, the state of our system – i.e., single versus multiple soliton - is solely dependent on the fixed pump power that needs to be set only once – a “set-and-forget” control parameter.

As stated above, the EDFA power is fixed, this is a highly useful control parameter that allows the *selection* of the final state. The system operation is independent of the turn-on dynamics. We show this in Fig. 1, and have revised the text accordingly:

‘The system consistently and repeatably starts up into the same desired state by simply setting the EDFA pump power to a fixed value, as shown in Fig. 1c-l. We consistently achieve the same single soliton state for an EDFA pump power of 320 mW. Figure 1c shows the microcomb output power, while Figures 1d,g show corresponding spectra and autocorrelation examples of the final state. Further, with the pump power set to 330 mW, the system consistently yields a two soliton state (Fig. 1h-l).’

Precisely to demonstrate that the operation state does not depend on the pump turn-on dynamics or timing, nor does it require the sweeping of any parameter, but depends only on the selected final power, we purposely varied the dynamics of the pump during the turn-on process while keeping the maximum power fixed, (Fig.2 d,e). This has never been achieved for microcombs. Remarkably, we obtain this for high energy solitons. This is totally different to dissipative Kerr solitons (DKS) that require precise pump sweeping dynamics- both in wavelength and power.

Another major advance of our approach is that it is intrinsically scalable in power, showing, at the same time, much higher efficiency than DKS states. We achieve over 10 mW (30 mW for multi-soliton states) of power in pulses that are background free. In other systems, there are always detrimental effects arising from thermal nonlinearities.

Once it is turned on, our system operates extremely stably and robustly without any external control at all. More importantly and quite remarkably, the same state naturally and rapidly re-emerges spontaneously, even if completely destroyed (typically requiring extreme perturbations). No other approach has ever achieved this.

Other methodologies, based on the sweeping of a parameter, or turn-key systems, require a specific turn-on point to get into the soliton state. They do not show this type of robustness.

The revised Fig. 3 demonstrates this capability specifically for single-soliton states that, as properly highlighted by the Referee, is a critical challenge. To highlight this point, we amended the text as follows:

‘Figure 3a shows that the soliton state consistently reappears even after strong system disruptions induced by external perturbations (Fig. 3a). The spectra in Fig. 3b, c show how the same soliton state is reliably recovered, [...]’

To highlight that we indeed recover the same state, we performed a highly accurate series of measurements of the microcomb frequencies, introduced as follows:

‘[...] with the comb lines offset within the microcavity resonances remaining constant to within a few per cent of the microcavity linewidth (Fig 3 d,e and f, g).’

The authors also state that the number of solitons is controlled by the loss of the cavity and accessible through control of the intracavity energy.

This is true, and it is an important feature of our system. We present two diagrams of states for one/two solitons and two/three soliton states in the supplementary section S2.1 and S2.2, respectively.

This laser-cavity soliton approach has been intensely studied by the authors and published elsewhere (e.g. Bao, et al., Laser cavity-soliton microcombs, Nat. Photonics 13, 284 (2019).) and theoretical analysis has also been performed in Cutrona, et al., Temporal cavity solitons in a laser-based microcomb: a path to a self-starting pulsed laser without saturable absorption, Opt. Express 29, 6629 (2021).

We thank the Referee for acknowledging our previous work. However, this statement is not correct. Regarding our work of 2019, we never reached single solitons states before. Furthermore, we did not demonstrate the ability of solitons to naturally form on their own, irrespective of the pump dynamics. Neither did we demonstrate the ability to spontaneously recover from complete disruption. Finally, our previous work did not discuss the role of the slow nonlinearities which are the keystone to the robustness and self-recovery of the soliton states that we achieve in our current work.

In fact, none of the five key achievements of our paper, listed at the beginning of this response, were achieved by any earlier work – either our own or any other published work.

In the field of microcombs, in general, nonlocal and slow nonlinearities are only now beginning to be explored - our holistic approach is fundamentally innovative in this field.

Finally, we stress that the theoretical analysis we present here was not performed in Opt. Express 29, 6629 (2021). That paper is entirely unrelated to this current work as it uses the same model as in our Nat. Photonics 13, 284 (2019) paper. The start-up of that system exhibited the same strong dependency on initial phases that are common to many other approaches, e.g. our citations Ref. [13,46-49]. Hence, our previous model published in Opt. Express 29, 6629 (2021) is incapable of predicting any of the results that we present here.

To clarify these key points, we revised the following passage in the Introduction:

'Nevertheless, all of these schemes presently require a specific system pre-configuration and the ability to execute a precise dynamical path towards the initiation of the desired soliton state. These strict and critical conditions - especially regarding the phase configuration - dramatically increase the system's susceptibility to external perturbations and, most importantly, do not offer any pathway for the soliton states to spontaneously recover.'

Summarising, here we propose a completely new model that includes slow nonlinear effects. The full details of the theory are presented in the Supplementary Materials. Thermal and gain effects that are crucial to our current achievements were not discussed in our previous work, including Opt. Express 29, 6629 (2021). In our current paper we explicitly include the equations describing the physics of these effects – this represents a major advance in this field. As the Referee can clearly see, the model that we propose here has four sets of coupled equations (two PDE's for the fields and two ODE's for the time-varying gain and thermal detuning). On the contrary, the theory presented previously, including in Opt. Express 29, 6629 (2021), comprises only the two PDE's for the fields. All of the new dynamics are explained thanks to the two additional ODE equations, modelling the thermal and the gain effects.

Thus, this work is too incremental and I cannot recommend this paper for publication in Nature.

We believe that our clarification and extensive set of experiments in answer to all the Referee's objections show clearly that our paper is a major contribution to this field. We have summarised in the introduction the five major breakthroughs that we conclusively achieve in this work – all of which solve key challenges for microcombs. In fact, those points include what the Referee states to be 'the major outstanding challenge' in the literature, which again is 'to get into the specific soliton state each time, particularly a single-soliton state.' We are grateful for the constructive criticism that has motivated us to more clearly highlight the breakthrough that we achieve in this work, to devise a new set of experiments which, in turn, have significantly helped us to improve the clarity and main message of our paper.

Specific comments are below.

1. The authors suggest that their approach is more robust as compared to the previously demonstrated approaches but the previous approaches consistently can achieve the same single soliton state based on their parameter settings. The present manuscript does not successfully achieve the same state implying that some configuration is also necessary to reach a certain single soliton or multi-soliton state. Can the authors clarify this point?

We are grateful for the comment since it gives us the opportunity to clarify this issue in more detail. Specifically, not only can we successfully achieve the same state, but we can also recover it and turn on the system from the off-state consistently into the same single soliton state. We have now provided a more straightforward demonstration of these features.

Figure 1 now shows a repeated start-up from the off state (below laser threshold) of the same single-soliton state, similarly as in Fig. 2 for different start-up trajectories. This is also the case when the state is destroyed via perturbations, and when a different soliton state (e.g. two soliton state) is selected. We demonstrate this by showing not only the spectral and temporal properties of the microcomb but also measuring the detuning of the microcomb lines from the microcavity resonances, which are consistently in the same position (within the tolerance of our system). All these features are now clearly displayed in the revised Fig. 3.

We have also revised the presentation of our original results (now Fig. 4) to illustrate better how the single and two soliton states are represented by fully separate and distinct regions in the diagram of states of parameters, and thus can be selected simply by setting the global cavity control parameter (i.e. the pump power).

2. It seems that, for a range of power and delay there's an existence range for the soliton state.

This is precisely the case and is the key feature of our approach. This enables the system to be set to operate in any chosen state simply by fixing these parameters once and for all – “set-and-forget”. This has never been achieved for cavity-solitons.

How much does the microresonator parameters affect the range? The generated combs are quite narrowband, does the existence range change for broader comb spectra? Can a broader comb be generated? How universal is the approach?

This is a very interesting point. Our system of equations is very general and universal, similar to the Lugiato-Lefever equations. Therefore, it can be applied to many different microresonator technologies.

Of course, increasing the optical bandwidth of the microcomb is always of interest and, as with any other system, our work leaves room for further developments. Engineering the waveguide dispersion of the microcavity, for instance, would increase the bandwidth further, similar to what has been achieved with the Lugiato Lefever systems before. While we plan to investigate this topic in a future work, we note that the microcavity dispersion is the parameter that predominantly determines the comb bandwidth, which is also true for other configurations, e.g. solitons in Ref [13], which are also well below 30 nm.

For our system, moreover, higher nonlinear regimes and broader combs can also be achieved by reducing the ratio between the microcavity linewidth and the free spectral range of the main laser cavity. This results in an increase in the bandwidth of the nonlinear state. In answer to the Referee's request, the new experiments were performed in a system where this ratio was reduced, which we achieved by decreasing the main laser cavity length. The Referee has noticed that in our previous results (now presented in the revised Fig. 4), we obtained microcombs with 30 nm bandwidth. They were generated with a laser cavity having a repetition rate of 77 MHz. In the revised Fig. 2, we show that larger bandwidth can be achieved with a cavity having a free spectral range of 95 MHz. With this configuration, we generated a single-soliton state with a significant optical bandwidth, well exceeding 50 nm, almost doubling the bandwidth we achieved before. (Please note: the measurements in Fig. 1 and 3 were performed with faster acquisition times due to the nature of the experiments being performed, resulting in a smaller dynamical range of about 30 dB, for our instrument. They nonetheless have a comparable spectrum).

In our approach, the thermal nonlinearity of the microresonator is particularly important, and, as discussed in the Supplementary Materials, our system can be engineered to meet different performance requirements by adjusting the balance between the slow nonlinearity in the main cavity and the EDFA pump power by simply varying the losses. This offers yet another critical degree of freedom in designing the system.

3. In Fig. 2C, middle plot, it seems that that this state may be a single-soliton state? Can the authors comment on the modulation in the spectrum that has a minimum near 1550 nm?

The Referee is indeed correct. In our original submission, there was already evidence of single-soliton state operation. The minimum of the spectrum in those spectra was due to a (spectrally) localised loss due to mode-crossing.

In this current work, we have included new experimental results that clearly and unequivocally show that we reliably and consistently achieve the single soliton state by choosing the system parameters judiciously.

4. In Fig. 5, the authors show the stability of the system by perturbing the pump power. How much of a perturbation is allowed for the system to return to the original Turing or soliton state?

We thank the Referee for raising this critical point.

Our system recovers not just from perturbations but from complete and total disruption – i.e., after being totally destroyed, which in our case required extreme measures such as banging the system physically with a screwdriver.

Furthermore, our new experimental results clearly and conclusively show that the system recovers to precisely the state it was operating in before the disruption, even after complete disruption or total erasure. Fig. 3 shows this conclusively where we focused on a single soliton state.

Note that we have moved the discussion around the old Fig. 5 into the Supplementary Material.

Referee #2 (Remarks to the Author):

Rowley et al. present a thorough experimental study, backed by numerical simulations, showing that a micro-cavity embedded into a simple fibre ring laser can lead to a self-starting cavity soliton operation. The cavity soliton terminology traditionally refers to a solitary pulse on background that is solution of the Lugiato-Lefever equation, which is not precisely the case in this work, as the generated solitons have more attributes of bright laser solitons. As a matter of fact, the authors emphasise on the advantage of these pulses with vanishing tails that allow a superior energy conversion ratio from the pump to the multiple comb lines, and I fully agree with them in that respect. Nevertheless, keeping the cavity soliton terminology for laser pulses brings some confusion. From the laser point of view, we rather have a situation of high-harmonic mode locking that is promoted by filter-driven four-wave mixing from an embedded micro-cavity. Otherwise, the fact that such laser cavity can achieve a self-starting harmonic mode locking without the assistance of an obvious saturable absorber mechanism is truly impressive.

We are grateful to the Referee for the in-depth reading of our current and previous works and for their appreciation of our results.

Indeed, we originally introduced the term “filter-driven four-wave mixing” in our 2012 Nature Communications paper to highlight that a Kerr microcavity embedded in a laser cavity was capable of mode-locking (harmonically) and could in fact produce coherent states in a fibre laser. As such, the Referee is correct in saying that this is the primary mechanism.

However, such a mechanism can also produce very different types of states, such as Turing states or solitons, both of which are coherent, mode-locked states but with deeply different physical natures. These states, see also [42], require a proper and distinct terminology. This was the motivation for introducing the new term “laser cavity-solitons” in our 2019 Nature Photonics paper, to which the Referee refers below, following the literature of cavity-solitons.

We are grateful to the Referee for highlighting the relevance of our earlier work to the pulsed laser community. Following their remark, we have added the following sentence in the Conclusions:

‘More generally, within the field of pulsed lasers, this work provides an effective approach to achieve self-starting, broadband pulsed laser without fast saturable absorbers that are notoriously difficult to realise, particularly for ultrashort pulses^{2,22}.’

The physical mechanisms involve a delicate combination of two slow (thermal and gain) nonlinearities to move the cavities resonances during the laser start-up phase and enable a soliton operation involving the fast Kerr nonlinearity and the chromatic dispersion.

The authors have an impressive expertise in such physical system and its modelling, well manifested for instance in their 2019 Nature Photonics article on “Laser cavity-soliton microcombs”.

However, I am not convinced that the present results would be suitable to Nature. Laser source performances are comparable to these of the 2019 paper.

We thank the Referee for the constructive criticism, which we have taken completely on board. We acknowledge that our previous presentation lacked clarity on this point. As further clarification, at the beginning of this response, we have listed the key achievements of this current work. None of these was reported in our 2019 Nature Photonics paper. Our results – and particularly now with the inclusion of our new experiments – conclusively demonstrate all of these key advances.

As we mention, we have added new experimental results that clearly show the control of single soliton states. In our 2019 paper we never observed single solitons.

We do acknowledge that our original experimental results for this submission did not clearly show that we reached the single soliton state. Now they do, on top of showing the clear, consistent, repeated, natural and spontaneous start-up into the same soliton state that can be easily controlled. We also now demonstrate conclusively the ability of our system to naturally recover to the same states spontaneously, even after complete disruption by extreme events. We demonstrate both the recovery and the long-term operation stability. The new figures (Figs. 1, 2 and 3) summarise this performance.

All of the key performance achievements that we present in this paper are new – based on fundamentally different physics, compared to our 2019 paper.

After a very good introduction on the fundamental challenges met by the control of cavity-soliton sources, the highlighted novelty is summarised in simple terms.

Then, the following of the manuscript becomes highly technical and not clearly written, requiring the reader to go back and forth with efforts from the figures to the text and the supplementary materials.

Among several possible examples, that of Figure 1c representing a numerical simulation which is not discussed, comes probably too early, and requires going through the supplementary material section.

We agree with the Referee that our presentation was very technical, and we have worked extensively on this point. The Referee will see that we widely clarified the main text, removed the simulations from Fig. 1 and, instead, devised a set of new experiments to discuss the start-up.

We have also completely redone all of the figures.

Basically, the manuscript is not entirely self-consistent and the interpretations of the figures are too scarce, while heaps of data are displayed on experimental maps. However, important experimental data is missing, such as detailed characterisation of the laser output in the time and radio-frequency domain for a selection of regimes (soliton state, Turing pattern...).

We have included new experimental results that allow for a more explicit discussion of our key advances.

We have redesigned the figures with a more explanatory approach. Hence, our first figure shows the evolution of the laser output at the start-up, the output spectrum and time measurement (autocorrelation) of the soliton data, as recommended by the Referee. Figure 2 illustrates the basic details of the new mechanism for soliton formation and stability. Here we present a laser scanning spectroscopy example. Such a figure should allow the reader to visualise this quantity that we will use in detail in the subsequent plots. The long-term characterisation of a stable state, including both the optical spectrum and radio-frequency noise suggested by the Referee, is now in Fig. 3.

In Figure 4 we have largely ‘unpacked’ the data contained in the maps, adding autocorrelation, spectra and detailed pots of the laser scanning spectroscopy that allows visualising the data that we synthesise in the state diagram. We did this for both CW, single and two soliton states. For the Turing patterns, their discussion is now in the Supplementary Material since they are not the central focus of this paper. More precisely, Supplementary section S2 summarises all the basic measurements (optical spectrum, autocorrelation, radio-frequency noise and laser scanning spectroscopy) for all the types of states that we discuss in the manuscript. We believe that this presentation is more suitable for a broad audience.

The supplementary material section provides numerous information but looks like a series of internal reports lacking fluidity.

We have also revised the Supplementary material, that now contains part of the material which we previously presented in the main text. As such, we have organised the text better with an introduction; we start with the theory, analyse the experimental measurements and then compare measurements with theory. The Supplementary Figures, although they do still present a large part of our original data, have been completely redesigned.

I also noticed dubious parameter values, such as $\beta_b \approx 60 \text{ ps}^2 \text{ km}^{-1}$ for the optical fibre and band-pass filter bandwidth $\Delta F_F = 650 \text{ THz}$, but I could be wrong of course.

Many thanks for highlighting this, $\Delta F_F = 650 \text{ THz}$ was indeed a typo, and we corrected it to $\Delta F_F = 650 \text{ GHz}$. Note that these values are now in a footnote, and we have also revised the theoretical presentation to add fluidity. For the dispersion, we need to consider that the fibre is mostly Erbium-doped in our setup. Hence the dispersion is different to standard SMF. To clarify this point, we added a reference to such a value in the manuscript (*Electron. Lett.* **27**, 1867 (1991)).

To conclude, I find the main text, past the introduction, needs a significant polishing. I find that the present manuscript is not, in its present form, suitable to the general readership of Nature but should rather be submitted to a Photonics journal.

We have taken the constructive criticism of the Referee(s) extremely seriously and have completely restructured the exposition of the subject, also performing further experiments that more clearly and directly show our 'truly impressive' results, making them accessible, we hope, to a broad audience. We are grateful for their appreciation of our work and their constructive criticism. We hope that, after this extensive revision, now the Referee will find the paper suitable for the general readership of Nature.

Referee #3 (Remarks to the Author):

The paper reports the development of a robust source of cavity solitons where a micro-resonator is used as a filter within a fibre laser cavity. If I understand correctly, the nonlinearities in the two systems (micro-resonator and laser cavity) act to balance each other, allowing for the stable formation of cavity solitons over a reasonably large parameter space. This, to me, seems to be a significant technological advance in terms of the development of systems for use in real-world applications. The results look convincing (well, at least those that I can understand) and I am not in any doubt that the system does what they claim. Where I am struggling a little is understanding some of the details of the results as I find the presentation quite technical for a general audience. In this regard, I have provided a series of comments that I feel would be helpful if the authors could address during their revisions.

We thank the Referee for the very positive assessment of our paper, their critical reading of the manuscript, their valuable suggestions, and for pointing out the significance of our work. We have substantially revised the paper in response and hope they will find our revised paper suitable for publication in Nature.

We agree that the presentation needed work, and we have significantly revised it, including simplifying it and completely redoing the figures that we acknowledge were previously quite technical. We believe that the figures are now much more accessible to a general audience.

In addition to this, we have performed further experiments that have enabled us to fill in a few gaps inherent to the experimental results reported in our original version.

We respond to the detailed and helpful comments point by point as follows.

1) The figures are very technical and the paper could really do with some schematics that help to illustrate the key points that would be understandable to a broad readership.

We have substantially revised the paper, including redoing the figures, in addition to including newly performed experimental results that complete some missing details in our original work. We believe that the revised paper makes our work much more accessible to a general readership (Fig.1-3 and Fig 4.). Following the Referee's suggestion, we added Fig. 2 which is a schematic (Fig. 2b) of the basic operational principle responsible for inducing locking in the system.

-For example, I cannot work out what Fig. 1c is trying to show – why is the plot inverted (red peaks are actually zero, blue background is maximum?) and what sort of round trip numbers are represented in the x and y axes (1 or many?). Does this even matter? I think the authors need to think clearly about what information they are trying to present here and whether it is accessible to readers without having to work too hard.

We agree with the Referee that the propagation dynamics were unclear. We redesigned Fig. 1, which now shows an experimental demonstration of the soliton start-up. We believe that this is more direct and better suited to a broad audience. Regarding the plot of our propagation, we took care to display them in the theory section, where all the quantities in the axis are now properly defined.

-Also, if I understand correctly, Fig 3 is supposed to show the flexibility of the system. However, this was lost on me the first two times I looked at this image. I wonder if there is a better way to highlight this? Maybe the insets could be brought out of the subfigures and made clearer? Or actually show some real data corresponding to points in the graph?

As mentioned, we have redone all of the figures, including the one referred to here, which is now Fig. 4. Following the suggestion of the Referee, we 'unpacked' the information of the state diagram. Figure 4 shows a subset of the figures, where we now show the real data that corresponds to a point in the state diagram. Specifically, we show the spectrum, autocorrelation and laser scanning spectroscopy, along with the plot of the

main quantities, such as microcavity line shift and frequency detuning of the oscillating lines within the microcavity resonance.

2) What are the key performance metrics for the soliton sources and what sort of applications could these find use in as the system is currently built? I.e., that are the power levels and/or pulse durations, and are these useful? And where is the output coupler? Is this in the fibre loop?

Cavity-soliton sources are presently an emerging technology. Robustness and stability are key fundamental goals and achieving these will be vital to enabling these devices to move beyond the laboratory into practical, real-world applications. We believe that the new figures illustrate the essential performance characteristics of our system much better than the original figures did. To further clarify this point, we have added an explanatory statement in the Introduction:

'A robust, repeatable approach for initiating and reliably maintaining the microcomb into the same type of soliton state, particularly the single soliton state, is widely acknowledged as being critical, with notable progress reported⁸⁻²⁰. Nonetheless, it largely remains the major outstanding challenge confronting this field.'

In general, our source has a large bandwidth (about 50 nm) and an output power of 10's of mW leading to a background-free pulse. Generally, this means that our spectral power density is about two orders of magnitude larger than the state of the art in this field. This makes our system particularly suitable for telecom and metrological applications where these types of combs are expected to be employed.

Regarding the output coupler, we have highlighted it in Figure 1a, showing that it is placed directly at the output of the microcavity. We further comment on this in the last point below.

3) Another key point that the authors stress is the robustness of the system. In this regard it would be helpful to know how robust the operation is to environmental disturbances and temperature fluctuations. In particular, does the dependency on the thermal nonlinearity in the micro-resonator place any limitations on the operation of the system? How long can the system remain in stable, and consistent, operation? Hours, days, longer?

We thank the Referee for this important remark. Our revised manuscript now includes new experiments that directly address these points.

Regarding the temperature variation and thermal nonlinearity of the resonator, from Fig. 4 and the Supplementary Section S2.1 we see that the single soliton region is maintained for shifts of the microcavity of about 5 GHz, (which corresponds to a change due to local heating of about 3°C). This resilience to temperature variations is about three orders of magnitude better than state-of-the-art results. For example, a remarkable performance in terms of temperature shift insensitivity was obtained in Ref. 15, where the authors showed resilience to thermal shifts in the order of a megahertz. On the other hand, we achieve a tolerance of 5 GHz, about three orders of magnitude larger. We redesigned Figure 4, with Fig.4c showing the microcavity lines that visibly shift while maintaining the laser lines locked on their red detuned slope.

Regarding the long-term stability, we now show in Fig. 3h-k a continuous measurement of optical and radio-frequency spectra of a single soliton state for about half an hour, which is limited presently by environmental factors. The performance of the system is actually better than this. A typical four-hour measurement under free-running operation with external disturbances is shown below. The Referee will see that the state is lost but recovers repeatedly. Here the state is subject to disturbances from the external environment (e.g. a typical university laboratory), which affects the system cyclically, inducing the state to be lost. Remarkably, however, the same state reappears.

Long term robustness of the state, about four hours continuous measurement of a single soliton state. Temporal evolution of the measured optical spectrum. The single state is lost and repeatedly recovered.

However, while we do demonstrate much greater stability and intrinsic robustness compared to any other system, we stress that exhaustive studies of reliability and environmental robustness are outside the scope of this paper, being more of an engineering issue. For this reason, we decided not to show the measurement above in the paper, which we include here just to answer the Referee's question. In the manuscript, we present results showing the proof of principle characterisation of the robustness of our system, including controlled measurements of the single soliton recovery in Fig. 3a-g. The data was obtained by literally hitting the system physically and repeatedly with a screwdriver. With each hit, the system was completely disrupted from operation (quite expectedly) but then, in every case, spontaneously recovered to the same state of operation without any external interference. This is unprecedented and is a significant advance over the state-of-the-art.

4) The system is sold as a robust, turnkey, source of cavity-solitons for practical applications, but how practical is the set up? The schematic in Fig 1a looks fairly compact and robust, but are the fibres really integrated with the chip? And what about the free-space section? How does it compare with the system published in Ref. [Nature 582, 365–369 (2020)] for size and practicality?

We thank the Referee for this comment. We have now added a picture that shows how the fibres are integrated with the chip in Fig. 1. Our chips are glued directly to a fibre array; hence, the system is very practical. Since we have a very high conversion efficiency (input power to the ring is well below 200 mW and we have a ~10 mW output), the limited power at the input of the ring allows us to use optical glue and make the setup very practical. To stress these features, we added the following sentence:

'Our samples use a glued fibre array directly on the chip (Fig. 1b), making the setup practical. Because our solitons have a very high conversion efficiency, the chip's input power is generally less than 100 mW (see Supplementary Section S2); hence we operate well below the damage threshold of standard optical glue.'

There is ample scope for improvement in the compactness and robustness of the physical setup. This, however, was not the focus of this initial work. The key point is that the microcomb source itself is integrated and fabricated with full CMOS technology. The fiber cavity can be readily integrated on-chip given the ultra-low losses that have been achieved in silicon nitride and Hydex waveguides. The free space section in the current system was included solely to conduct the experiments more easily, but in prototypes, this would be eliminated by designing the cavity to the correct length. There are many other easy modifications that would improve the compactness. These sorts of issues are very much the task of standard engineering. Existing custom EDFAs can be also made extremely compact.

We believe that our approach does not have any significant disadvantages compared to other work (ie., Nature 582, 365–369 (2020)), and on the other hand has many advantages that we have outlined, including the ability to achieve much higher power operation, compared to typically a few hundred microwatts [Nature 562, 401–408 (2018)] where post-amplification is needed for practical use.

The ability of our system to operate at much higher powers is useful for many applications. We believe that our effective use of the thermal nonlinearities to form the dominant attractor will be useful also in other approaches. We have added:

'We achieve this by transforming the soliton states into dominant attractors of the system, and experimentally demonstrate this approach in a microresonator-filtered fibre laser. This method is fundamental and very general, applicable to many systems, particularly those based on dual-cavity configurations such as self-injection locking^{8,9,13}.'

5) I normally expect to be able to find technical details (fibre lengths, resonator size, pump wavelength and power levels etc.) somewhere in the paper or supplementary information. Maybe I missed it, but this information was not obvious to me.

We have added this information to the methods:

'The experimental setup consists of nesting a high index doped silica^{21,50,51}, integrated ring-resonator, with a free-spectral range of ~48.9 GHz, a 1.3 million Q-factor, and a positive (focusing) Kerr nonlinearity of about 200 times that of silica, in an add-drop configuration into an amplifying, polarisation-maintaining, fibre cavity. Our samples use a glued fibre array directly on the chip (Fig. 1b), making the setup practical. Because our solitons have a very high conversion efficiency, the chip's input power is generally less than 100 mW (see Supplementary Section S2); hence we operate well below the damage threshold of standard optical glue.'

For Figs. 1-3 we used a fibre cavity with a free-spectral range of ~95 MHz and a microcavity sample with linewidth <120 MHz. The results of Fig. 4 are for a longer cavity with free-spectral range ~77 MHz and a microcavity sample with linewidth <140 MHz. We used two different microresonators with similar properties. In both cases, the fibre cavity includes a ~1-meter polarisation maintaining optical amplifier and a free-space section containing a motorised delay line, polarisation control optics to govern the cavity losses, and a 12 nm wide bandpass filter.'

This information is now also in the introduction:

'Figure 1 shows a simple embodiment of this approach based on a microresonator-filtered fibre laser^{21,50,51} (Fig. 1a). An integrated microring resonator (Fig. 1b, free-spectral range, FSR=49.8 GHz) is nested within an Erbium-doped fibre amplifier (EDFA) lasing cavity. Here we use a ~ 2 m fibre loop with an optical path set, approximately, to a multiple of the microcavity length within a tolerance of a few hundred microns (FSR=95 MHz).'

We have now rewritten a large part of the manuscript, and we also added to the Fig.1 caption:

'Microcomb laser: the nonlinear Kerr microresonator (FSR = 48.9 GHz) completes the fibre laser cavity (FSR = 95 MHz). The global cavity controls are highlighted: a section containing the variable EDFA 980 nm pump, an optical filter, polarization controls and a delay stage to approximately match the repetition rate of the fibre cavity with a submultiple of the microcavity FSR. The fibre-coupled output port of the microresonator is highlighted.'

Reviewer Reports on the First Revision:

Referees' comments:

Referee #1 (Remarks to the Author):

The authors show spontaneous pulse formation using a microresonator-based filtered fiber laser system. In my previous report, one of the key issues I raised was that it is important in the community that a specific pulse state is reached repeatedly. I appreciate the authors' effort to address this thoroughly with a combination of experimental and theoretical work. However, the claim of a single soliton in their revised manuscript raises more questions than answers and I cannot recommend the publication of this manuscript.

In the previous work on microcombs operating in the anomalous dispersion regime, the state that is excited corresponds to a dissipative Kerr soliton which is characterized by a sech² spectral profile. The spectral profile shown in this work, particularly that of Fig.2e is quite far from such a spectral profile. In their response, they have mentioned that the spectral bandwidth of their comb output is dependent on the dispersion of the microresonator. However, all of their spectra seem to be confined to the gain bandwidth of their erbium-doped fiber. It would be more convincing to demonstrate a spectrum that spans beyond the gain bandwidth of an erbium-doped fiber. Alternatively, adding numerical modeling with a different dispersion profile would also provide insight into how the dispersion affects the generated comb spectrum.

In addition, more explanation can be provided for how the multiple peaks in the "pump" mode affect the spectrum, the local minimum in their spectrum in Fig.1d and f, the reasons for the large modulations in their spectra in Fig. 2.

The authors also promote their high power and high efficiency. However, in their system, they start with 330 mW of pump power to generate a 10 mW output. While it is hard to quantify the conversion from the pump modes within the comb to the 'soliton' state in this work, 3% conversion efficiency from their pump to comb can hardly be considered efficient and high power output. In fact, there have been numerous work discussion the efficiency of dissipative Kerr soliton formation and previous analysis has already shown that it is indeed possible to achieve Kerr combs with efficiencies beyond 20%. For example, please see Gartner, et al., Phys. Rev. A 100, 033819 (2019).

Referee #2 (Remarks to the Author):

The authors have gone through a major overhaul of their manuscript, which was essential and requested by all referees. The new manuscript has become pleasant to read, the main text nearly self-consistent while having a strong backing from organized supplementary materials. The physics of the nonlinear laser-cavity system finally reveals itself quite clearly.

Now, the key question: does the manuscript, through its estimated novelty and potential scope, merit publication in Nature? I have been pondering this question. The physical system, associated experimental setup, and pulsed operation regimes are quite close to those presented in some earlier publications, as noticed by referees. However, the authors reach here a new and deep understanding of the mechanism allowing the self-starting, stabilization, and potential recovery of energy-efficient soliton pulse trains at high repetition rates – and their frequency comb counterpart in the spectral domain. This mechanism involves a subtle balance between the slow thermally induced nonlinearities from both microcavity and doped-fiber cavity. In short, the conceptual novelty can be summarized by the understanding of figures 2a-c, now clearly explained in the text, and supported by a numerical model (Suppl. Material) that is indeed a novel and major

extension to previous models. With the present manuscript and the authors' response, nearly all my doubts and questions have been answered. By understanding the complex attracting state with slow and nonlocal nonlinearities, the laser-cavity more than ever represents an interesting alternative to driven microresonator systems for frequency comb generation.

Therefore, I will now support publication in Nature of the manuscript, suggesting that the authors clarify the remaining following few points.

1) Title: following a more common terminology in pattern formation, I suggest the title "Self-Emergence of Robust Micro Cavity-Solitons" instead of "Natural Emergence of Robust Micro Cavity-Solitons". I don't feel comfortable with the qualifier "natural" in this physical context.

2) Long pumping power ramp and Figure 2.

a) It now makes clear that the slow ramp is used for illustrative characterization purposes. However, a slow ramp reduces instabilities in the pattern forming stage. Can it be much faster, say around 1 second duration or shorter and still reach the desired soliton state?

b) The trajectory represented in Fig2a starts from a point in the high gain region. That does not correspond to actual start-up trajectories, initiated from a pump off. What happens when the pumping power ramp is varied? Does the trajectory enter the stable soliton region in Fig 2a earlier (at lower gain), followed by an adiabatic evolution? This seems in contradiction with the large fluctuations seen during the ramp in fig 2d. Or maybe Fig2a is just too schematic?

3) For some reason still unclear, the 2-soliton state does not yield a symmetric time distribution. Yet, the distribution seems to equalize with an increased pumping, as can be evidenced from figure 4a, upper autocorrelation trace. Can the authors elaborate on that?

4) Figure 3jk: why taking the RF of the fundamental frequency of the long cavity, whereas it is operated at high harmonics? To discuss stability of the pulsed regime, it would be more instructive to take the RF spectrum at 48.9 GHz. In the text, the authors indicate an ultra-low noise radio-frequency spectrum, but this is not really substantiated, with a SNR limited to around 20dB.

5) From Fig 4c we see that the needed thermal redshift of the main laser cavity exceeds just a little that of the microcavity. Could the author explain better how this is not just good luck? Indeed, as the authors write, nonlinearities of the same focusing type nearly cancel each other. Would other materials for the nonlinear microcavity be suitable? How to design both cavities to routinely benefit from that situation? Is playing with lumped losses a practical or even a required solution (I acknowledge the important consideration about these losses in the suppl material section)? It is important to better assess the practical reach of the authors' findings in that respect.

Author Rebuttals to First Revision:

Referees' comments:

Referee #1 (Remarks to the Author):

The authors show spontaneous pulse formation using a microresonator-based filtered fiber laser system. In my previous report, one of the key issues I raised was that it is important in the community that a specific pulse state is reached repeatedly. I appreciate the authors' effort to address this thoroughly with a combination of experimental and theoretical work.

We appreciate the Reviewer's acknowledgement that we have indeed achieved a clear demonstration of the repeatable and natural (self-emergent) generation of specific desired soliton pulsed states, being this was raised as an issue of concern in the previous round of reviews.

However, the claim of a single soliton in their revised manuscript raises more questions than answers and I cannot recommend the publication of this manuscript.

While we believe that we had already presented experimental results that clearly demonstrated the generation of single soliton states, including clear experimental proof of single pulse temporal autocorrelation, we are nonetheless committed to fully clarifying this point. We agree with the Reviewer that this issue is a central one.

In response to the previous comments, we performed five additional experiments. **We use these new data to revise the graphs that were shown in Figs. 1-3. These new data now clearly eliminate and resolve one of the key features that the Reviewer had issues with – i.e., a small coexisting CW perturbation. Furthermore, we have added theoretical calculations (also shown) of single soliton states that match our experimental spectra extremely well (see below). We have added all of this to the paper in a new Extended Data Fig. E1. Overall, these new data ultimately resolve all the issues raised by the Reviewer.**

- 1) In the previous work on microcombs operating in the anomalous dispersion regime, the state that is excited corresponds to a dissipative Kerr soliton which is characterised by a sech^2 spectral profile. The spectral profile shown in this work, particularly that of Fig.2e is quite far from such a spectral profile.

The Reviewer is correct in saying that our spectral profiles diverge from the shape of a sech^2 intensity profile. In fact, we had already addressed this feature in the past – discussing the characteristic of our soliton shapes in substantial detail in our 2019 Nature Photonics paper. We briefly summarise that discussion here.

Different types of equations yield different types of soliton profiles – the sech^2 profile is only one example.

A sech^2 intensity profile is the solution of a soliton in the Nonlinear Schrödinger equation, which, under certain conditions, can serve as a first approximation in driven lossy systems such as, for example, in the Lugiato-Lefever equation (LLE). Also, for the LLE and hence, for the dissipative Kerr solitons, a sech^2 intensity profile is only an approximation. The soliton solution of the LLE can depart from this, especially in the tails. [see I. V. Barashenkov and Yu. S. Smirnov, "Existence and stability chart for the ac-driven, damped nonlinear Schrödinger solitons," Phys. Rev. E 54, 5707, 1996; Stéphane Coen and Miro Erkintalo, "Universal scaling laws of Kerr frequency combs," Opt. Lett. 38, 1790-1792 (2013)].

Indeed, it has been widely reported and acknowledged in the literature that in both conservative and dissipative systems, soliton profiles in general do differ from the sech^2 profile [see “Dissipative solitons: from optics to biology and medicine,” Ed. A. Ankiewicz, and N. Akhmediev, (Springer, 2008); T. Ackemann, W. Firth, and G.-L. Oppo, “Fundamentals and applications of spatial dissipative solitons in photonic devices,” *Adv. At. Mol. Opt. Phys.* **57**, 323 (2009); “Nonlinear optical cavity dynamics: from microresonators to fibre lasers,” Ed. P. Grelu (Wiley, 2015)]. It is typical, moreover, that systems having coupled equations diverge significantly from the solutions for the two separate independent equations, a very famous case being the coupled nonlinear Schrödinger equation [P.G. Kevrekidis, and D.J. Frantzeskakis, “Solitons in coupled NLS models: A survey of recent developments,” *Review in Physics* **1**, 140 (2016)], which have a much broader set of solutions than merely sech^2 .

The important point here is that the fact that our profiles differ from a sech^2 profile does not in any way affect the bright soliton nature of the solutions presented in our paper.

We had already demonstrated in our Refs. [21] and [48] that laser cavity-solitons generated in double cavities have a spectral profile that diverges from the solution to the nonlinear Schrödinger equation. In our previous theoretical paper, Ref. [48], (we stress again that these studies did not address in any way the self-emergent or recoverable operation that we report here and fully model in the Supplementary) we have clearly shown how our different types of solutions actually evolve from a sech^2 profile, obtained for a system with a purely linear gain equation. The spectral shape changes due to the presence of group velocity dispersion and gain dispersion in the amplifying cavity. We discuss this point in Comment #6, along with the spectral behaviour at the through port and its difference with the drop port.

- 2) In their response, they have mentioned that the spectral bandwidth of their comb output is dependent on the dispersion of the microresonator. However, all of their spectra seem to be confined to the gain bandwidth of their erbium-doped fiber. It would be more convincing to demonstrate a spectrum that spans beyond the gain bandwidth of an erbium-doped fiber.

We respectfully yet firmly disagree with this. Our spectrum, in fact, does exceed the gain bandwidth of our laser, and this was already demonstrated and discussed in our previous work, i.e. Ref. [21]. As we report in the methods for the current work, the gain in the cavity is actually limited by a 12 nm passband filter, not the EDFA bandwidth. As we discuss in the response to Referee 2, this filter is important because we need to select the red tail of the Erbium gain to induce the correct nonlinearity for the locking to take place. In order to definitively prove this point, we now include clear, direct measurements of the gain profile under pumping, superimposed with the soliton spectral profile In Fig. E1. As shown below in Comment #6, it is very clear that the soliton spectrum exceeds the amplification bandwidth by about 40 nm for our broader case. We have revised the manuscript accordingly:

In the methods, we introduced a new section ‘Characterisation of the soliton spectra and numerical fitting’

The numerical fit of a typical experimental spectrum of this output is reported in Fig. E1 a,b and d,e for the soliton spectra in Fig.1 c and Fig.2 f. Here, we also present the experimental measurement of the gain+12 nm filter bandwidth, showing how the soliton spectrum well exceeds the amplification spectrum.

- 3) Alternatively, adding numerical modelling with a different dispersion profile would also provide insight into how the dispersion affects the generated comb spectrum.

We agree and have done this, as discussed below.

In addition, more explanation can be provided for

- 4) how the multiple peaks in the “pump” mode affect the spectrum,

We do not understand what the Reviewer refers to as the ‘pump mode’, given that ours is not a CW pumped system. We believe that the Reviewer is referring to the peaks at 1538 nm, where a maximum of the Erbium gain is located. We have discussed this already at length in the Supplementary Section S2, where we clearly show that these peaks contain both red detuned modes, belonging to the soliton, and an independent, blue detuned CW line. We had commented on this previously:

‘The soliton states have red-detuned oscillating lines. High energy soliton states, appearing at high EDFA pump powers, are entirely red-detuned. Interestingly, some single soliton cases show the coexistence with a few blue-detuned modes close to 1540 nm. This coexistence occurs in a spectral region where the system’s gain has a strong and narrow peak. There, we observe a local maximum of the amplification due to the combined effect of the gain shape and intracavity spectral filtering. In this region, the EDFA exhibits a substantial dispersion of the nonlinear refractive index, which also decreases. We attribute to such two effects the presence of those modes superimposed to the red-detuned soliton modes.’

These modes do not affect the soliton spectrum because they are detuned and independent, as already discussed in our paper Ref. [21].

Nonetheless, to be definitive and for the sake of clarity, ***we repeated all of the measurements of the start-up for the single solitons to show clear examples of CW perturbations with a much-reduced amplitude, which are essentially negligible. A typical case is reported below in the new Fig. 1***

Figure 1. Natural onset of cavity-solitons. **a** Microcomb laser: the nonlinear Kerr microresonator (FSR = 48.9 GHz) completes the fibre laser cavity (FSR = 95 MHz). The global cavity controls are highlighted: a section containing the variable EDFA 980 nm pump, an optical filter, polarisation controls and a delay stage to approximately match the repetition rate of the fibre cavity with a submultiple of the microcavity FSR. The fibre-coupled output ports of the microresonator are highlighted. **b** Picture of the microresonator photonic-chip with integrated fibre coupling. **c** Repeatable start-up of the same single soliton state from the off state, a temporal measurement of the microcomb output power (blue line), stabilising to 4 mW. The EDFA pump power (green line) is increased from 0 mW to 350 mW in 2 s. **d, e** Output spectrum and autocorrelation of the microcomb after the first start-up, at 30s. $\Delta T=20$ ps is the time period corresponding to one round-trip of the microcavity. **f, g** Same for the microcomb emitted after the second start-up, at 95s. **h-l** Same as c-g, for a two soliton state, selected by driving the EDFA at a higher regime power of 380 mW.

We feel that we have now fully clarified this point.

5) the local minimum in their spectrum in Fig.1d and f,

The Reviewer had already asked about the local minimum in the previous round of reviews, and we had already answered that it is due to losses arising from mode crossing. For the sake of clarity, we plot here the linear transmission of the ring.

ASE transmission of the EDFA with 12 nm filter and microring: the dip centred at 1548 nm is a linear loss due to local mode-crossing. The peak around 1538 nm is due to the ASE profile.

These losses are intrinsic to the linear behaviour of the sample and not due to the nonlinear interaction. These are indeed minor and have essentially no impact on the system operation. In fact, this has already been discussed in our previous paper Ref. [21] (see Figure 3 from Ref. [21] also reported in Comment #6 below). These features are more visible in the profile acquired at the through port due to interference.

6) the reasons for the large modulations in their spectra in Fig. 2.

We have already discussed the modulation of the spectrum at length in our previous paper Ref. [21]. We summarise this again here. Our spectrum can depart from the sech shape of the nonlinear Schrödinger equation. For the set of parameters in our current experiments, this solution develops into the modulated solution that we observe here. This can be clearly seen in Fig. 2c of our Nature Photonics paper Ref. [21], which we report here below for clarity. In Ref. [21] we had already developed the theory to show that these features arise in bright soliton states, where the theory accurately fits the experimental spectra. We highlight a key passage in the caption – “Note that solutions at lower energy possess instead a profile that is more closely related to a sech^2 profile.” We stress again that that theory was only aimed to fit the fast features - e.g. the spectral features - of the system and not the slow start-up dynamics that we demonstrate and focus on in this current paper (see also Supplementary).

It is clear that the soliton pair observed in Ref. [21], Fig. 3, is of a similar nature to the single soliton reported in our current manuscript, and this was the reason that we did not highlight this point here since we assumed that this was obvious. The fact that such a profile corresponds to a bright pulsed solution was already clearly demonstrated in Ref. [21].

Nonetheless, we do acknowledge that in our Ref. [21] we had consistently shown only the output at the drop port of our microcavity, which reproduces the microcavity field, while in previous versions of our current paper we had presented only the through port in Figs. 1-3, which shows the effect of the interference between the microcavity and the amplifier-cavity fields resulting in more pronounced spectral dips. This output was not theoretically fit in any of our previous works. For the sake of clarity, we have now added in Fig. E1 the spectral characterisation of the single soliton states reported in our Figs. 1 and 2 for both the drop-port and through-port outputs, along with the corresponding numerical fitting.

Figure E1 Characterisation of the soliton state **a-c** Example soliton state, obtained with a 350 mW pump at 980 nm. Experimentally obtained optical spectra of a single soliton (blue) with their theoretical fit (red dashed) at the **a** ‘through’ and **b** ‘drop’ ports. In the fit we use $g=0.08$ and $\Delta=0.46$, with the other parameters reported in the Methods. The operational EDFA gain bandwidth is indicated by the overlay in yellow. The input power to the microchip is 44 mW and the measured output powers are 4 mW at the ‘through’, and 6.5 mW in the intracavity ‘drop’ ports, respectively. The insets correspond to the autocorrelation trace, clearly showing one peak per microcavity round-trip. **c** Three examples of the intracavity spectrum (blue), displaying the lasing modes within each microcavity resonance and corresponding to the highlighted wavelengths. While the modes are all red detuned, we observe in the mode around 1538 nm, where the peak of the Erbium is located, the coexistence of the soliton red-detuned mode with a blue-detuned continuous wave mode. **d-f** Same as **a-c**, for a single soliton with a slightly higher pump power of 360 mW at 980 nm. In the fit we use $g=0.1$ and $\Delta=0.4$, with the other parameters reported in the Methods. The input power to the microchip is 63 mW and the measured output powers are 5 mW at the ‘through’, and 8.3 mW at the intracavity ‘drop’ ports. **e** Same as **c**, for the spectrum shown in **d**.

As can be seen, the theory and experimental results are in excellent agreement and feature a single soliton state with only second-order anomalous dispersion present in both cavities. The effect of higher-order dispersion in the microresonator is negligible in our configuration. The spectra at the drop port now show that the field in the resonator is smooth and has a sech-like profile for low energy cases. Two regions (about 1535 nm and 1550 nm, at the edge of the amplification spectrum) with some spectral depletion are visible in the through port. These features do not affect the overall bandwidth or high quality of the spectra and are due to the interference between the optical field at the input of the resonator and the field circulating in the microcavity. Also, these spectra are straightforwardly fit by the model. The figure also reports clear evidence that all our spectra substantially exceed our amplification bandwidth (yellow line). Although the Erbium gain is clearly very far from being flat, varying by almost 10 dB over the amplifier bandwidth, our pulses are nonetheless remarkably symmetrical; the only feature that appears due to the large amplification bump around 1538 nm is the small presence of the CW perturbation mode discussed above. We stress again that we have largely reduced the presence of such perturbation which is very minor in this set of experiments. For clarity, we added in Fig. E1 additional proof that this mode is independent. The resonances reported in Fig. E1 c and f clearly show the presence of this perturbation as an ‘additional’ blue detuned mode, well distinguished from the soliton red detuned modes present along all the spectra.

We added also a discussion in the methods:

A lossy nonlinear Schrödinger equation models the evolution of the variable a for the microcavity field in the time and space coordinates t and x , normalised respectively to the fibre cavity roundtrip and microcavity length. The field in the main amplifying loop is b_0 and represents the leading supermode (i.e. the set of modes filtered by the microcavity). A generic supermode is represented by the field b_q

$$\partial_t a = \frac{i\zeta_a}{2} \partial_{xx} a + i |a|^2 a - \kappa a + \sqrt{\kappa} \sum_{q=-N}^N b_q, \quad (1)$$

$$\partial_t b_q = \frac{i\zeta_b}{2} \partial_{xx} b_q + \sigma_6 \partial_{6x} b_q + 2\pi i (\Delta - q) b_q + g b_q - \sum_{p=-N}^N b_p + \sqrt{\kappa} a. \quad (2)$$

Here ∂_{xx} and ∂_{6x} are a second and sixth order derivatives. The parameter Δ represents the normalised frequency detuning between the two cavities, g is the normalised gain while the group-velocity-dispersion coefficients are $\zeta_{a,b}$, with values of $\zeta_a = 1.25 \times 10^{-4}$ and $\zeta_b = 2.5 \times 10^{-4}$. As the gain is taylorled with a 12 nm flat-top filter, we use a sixth order derivative to reproduce the gain dispersion, with $\sigma_6 = (1.5 \times 10^{-4})^3$. The coupling coefficient is $\kappa = 1.5\pi$. Further details are reported in Supplementary section S2. These parameters are used to fit the experimental spectra in the Extended Fig. E1 with a numerical mode-solver that provides the nonlinear eigenmodes of the system, including the soliton functions, as in Refs^{3,9,10}. In particular, the field measured at the ‘drop’ port directly reports the microcavity internal field a . The field at the output port, which is the ‘through’ port of the microcavity, can be theoretically evaluated with $c(t) \approx b_0 - \sqrt{\kappa} a$. To fit the experimental data, we included an additional component αb_0 to this value, where α is a coefficient that accounts for the non-ideal transmission of the microcavity and the polarisation interference at resonance. The numerical fit of a typical experimental spectrum of this output is reported in Fig. E1 a,b and d,e for the soliton spectra in Fig. 1 d and Fig. 2 e, respectively. Here, we also present the experimental measurement of the Erbium gain+12 nm filter bandwidth, showing how the soliton spectrum well exceeds the amplification spectrum.

To summarise, we have provided clear evidence that all of the spectral discrepancies commented on by the Reviewer are actually very minor effects, and indeed many have already been discussed at length in our previous papers (as well as in papers from other groups). They do not affect the nature of our bright soliton spectra. These bright solitons were already demonstrated in our Nature Photonics paper Ref. [21]. We emphasize that none of these remarks affects, in any way, our claim of generating single solitons. This is definitively proven by the clear and unambiguous autocorrelation measurements that we show in all the experimental sets reported in our paper (see also the Extended Data Figure E1). We thus feel that we have completely resolved this issue. We appreciate these comments that gave us the opportunity to clarify these points. We feel that they have improved the overall quality of our data and clarity of our presentation.

- 7) The authors also promote their high power and high efficiency. However, in their system, they start with 330 mW of pump power to generate a 10 mW output. While it is hard to quantify the conversion from the pump modes within the comb to the ‘soliton’ state in this work, 3% conversion efficiency from their pump to comb can hardly be considered efficient and high power output. In fact, there have been numerous work discussion the efficiency of dissipative Kerr soliton formation and previous analysis has already shown that it is indeed possible to achieve Kerr combs with efficiencies beyond 20%. For example, please see Gartner, et al., Phys. Rev. A 100, 033819 (2019).

Of course, we agree that efficiency is an absolutely key issue, but we respectfully yet firmly disagree with these comments.

The paper cited by the Reviewer is indeed a significant *theoretical* study which proves that the conversion of Kerr bright solitons scales with the repetition rate of the microcavity and the pump power. This is discussed in a number of places including in K. Jang, Yoshitomo Okawachi, Yun Zhao, Xingchen Ji, Chaitanya Joshi, Michal Lipson, and Alexander L. Gaeta, "Conversion efficiency of soliton Kerr combs," *Opt. Lett.* **46**, 3657-3660 (2021).

That work, however, did not show that a 20% efficiency is widely achievable, but rather that for most practical conditions, efficiencies are in fact much lower. As clearly discussed also in (*Opt. Lett.* **46**, 3657-3660 (2021)) this theory points out that the large resonators that are needed to achieve microwave compatible frequency spacings (<50GHz) suffer from highly inefficient conversion, and particularly at pump powers that are compatible with integrated photonics.

Indeed, those authors explicitly showed that 20% efficiencies are achievable in experimentally feasible conditions only for very small resonators, with extremely high (about 1THz) repetition rates, as well as in the highly over-coupled regime. For a 50 GHz resonator, as the one we use, this value drops to well below 1% even in the best-case scenario, as was clearly stated also by Jang et al., [*Opt. Lett.* **46**, 3657-3660 (2021)] *'Furthermore, combs with sufficiently low FSRs to be detected electronically will operate with efficiencies $\leq 1\%$, which could pose challenges to implementing low-power self-referenced chip-based Kerr combs.'*

In our view, the Reviewer is in fact not making an equitable comparison. Our experimental results are on the *experimental* conversion from a CW 980 nm pump laser to a C band microcomb. Hence, this involves *both laser energy conversion and nonlinear conversion*. On the other hand, that *theoretical paper* considered *only* the *nonlinear conversion* of a pump to a microcomb having the same optical frequency carrier, without accounting for the laser conversion. An Erbium amplifier has an intrinsic laser conversion value of 30%. With reference to our latest results, for a through output power of 4 mW, the input to the microcavity (the pump in the C-Band) is about 44 mW, and it is this number that needs to be used in the comparison, not the one that the Reviewer uses.

Finally, we point out that our work employs a 4 port device. This is rather uncommon in the microcomb community that typically uses only 2 port devices. The use of a 4 port device provides an additional output that needs to be accounted in the efficiency. The power at the drop port in our experiments is 6.5 mW. The chip coupling loss is 1.5 dB at each port. Summing up the on-chip conversion of both ports we achieve a nonlinear experimental *on chip* conversion of about 50% - which is actually the correct number that should be used for comparison, with ~6.5 mW and ~9.3 mW on-chip at the two ports, when pumping with about 30 mW on-chip. These values, moreover, were obtained using a micro-ring resonator that has an FSR in the microwave compatible region, i.e. a repetition rate of 50 GHz. We have added the following text to the Methods section to give more experimental details.

For the states in Fig. E1 a,b and d,e, the input powers to the microcavity were 44 and 63 mW, respectively. The 'through' output powers were instead 4 mW and 5 mW. The second output ('drop port') was reconnected to the amplifier leading to off-chip emitted powers of 6.5 mW and 8.3 mW, respectively. Part of the light was extracted for characterisation with a beam splitter. The total cavity operated with ~10 dB gain. When accounting for the on-chip losses of 3 dB, we estimated that the microresonator operated with 31 mW and 44 mW on-chip input powers. The on-chip 'through' output powers were 5.7 mW and 7.1 mW, while the on-chip 'drop' port powers were 9.3 and 11.8 mW. This results in an on-chip nonlinear conversion efficiency of about 20 and 30 % at the 'through' and 'drop' ports, respectively.

Referee #2 (Remarks to the Author):

The authors have gone through a major overhaul of their manuscript, which was essential and requested by all referees. The new manuscript has become pleasant to read, the main text nearly self-consistent while having a strong backing from organised supplementary materials. The physics of the nonlinear laser-cavity system finally reveals itself quite clearly.

We are very pleased and inspired by the fact that the Reviewer is now in a confident position to positively evaluate our work. We believe that their comments and criticism have been essential in improving the readability and structure of our paper.

Now, the key question: does the manuscript, through its estimated novelty and potential scope, merit publication in Nature? I have been pondering this question. The physical system, associated experimental setup, and pulsed operation regimes are quite close to those presented in some earlier publications, as noticed by referees. However, the authors reach here a new and deep understanding of the mechanism allowing the self-starting, stabilisation, and potential recovery of energy-efficient soliton pulse trains at high repetition rates – and their frequency comb counterpart in the spectral domain. This mechanism involves a subtle balance between the slow thermally induced nonlinearities from both microcavity and doped-fiber cavity. In short, the conceptual novelty can be summarised by the understanding of figures 2a-c, now clearly explained in the text, and supported by a numerical model (Suppl. Material) that is indeed a novel and major extension to previous models. With the present manuscript and the authors' response, nearly all my doubts and questions have been answered. By understanding the complex attracting state with slow and nonlocal nonlinearities, the laser-cavity more than ever represents an interesting alternative to driven microresonator systems for frequency comb generation. Therefore, I will now support publication in Nature of the manuscript, suggesting that the authors clarify the remaining following few points.

We thank the Reviewer for their support, we will answer the remaining few points below.

1) Title: following a more common terminology in pattern formation, I suggest the title “Self-Emergence of Robust Micro Cavity-Solitons” instead of “Natural Emergence of Robust Micro Cavity-Solitons”. I don't feel comfortable with the qualifier “natural” in this physical context.

Thanks for the valuable suggestion, we have revised the title as suggested.

2) Long pumping power ramp and Figure 2.

a) It now makes clear that the slow ramp is used for illustrative characterisation purposes. However, a slow ramp reduces instabilities in the pattern forming stage. Can it be much faster, say around 1 second duration or shorter and still reach the desired soliton state?

The answer is yes - the system responds to faster ramps of about 1s similarly to ramps of about 5s. In fact, our experimental results confirm that a faster ramp does not offer any particularly new insights or significant changes to the dynamics. Indeed, this was the main reason we did not originally include these results.

We need to consider that we are using a bulk commercial amplifier, that the system is not explicitly environmentally controlled and that we are observing its typical timescales when starting from cold. In the new experiments conducted for this latest revision, *we have repeated the set of*

measurements with faster ramps, and we took care to include an example with a 1s ramp, to better emphasise this point.

Figure 2. Principle of operation for the natural onset and intrinsic stability of cavity-solitons. **a** Diagram of states for a microcomb laser. Here the coordinates are two typical parameters, frequency detuning (x-axis, here scaled to our experimental setting) and gain (y-axis). The gain roughly correlates with the EDFA pump power, further details are in the Supplementary Materials. The start-up region is in blue. The stable solitons (orange) region is well within the no start-up (white) region. In our system, the soliton behaves as a dominant attractor (dark-blue path). Note that the regions with different soliton numbers are perfectly superimposed here, further details are in the Supplementary. **b** Microcavity resonance (purple) and laser modes (red) during stable soliton operation. The energy-dependent red-shift of the laser modes is greater than that of the microcavity. As such, the system preferentially locks to the laser mode red-detuned to the microcavity resonance. The orange arrows highlight the frequency detuning parameter, defined as the difference between the microcavity central resonance and the laser mode. **c** Laser scanning spectroscopy of a microcavity resonance (bandwidth 120 MHz, Q-Factor of 10^6) under lasing condition. The red-detuned lasing frequency is visible as a sharp peak highlighted by a red arrow. **d** Experimental start-up of a single soliton from the off state. Microcomb output power vs time (blue) and EDFA pump power (green). The EDFA pump is ramped from 0 mW to 360 mW. The three panels indicate different ramp times of 1 s, 5 s and 10 s, respectively. **e** Experimental output spectra and autocorrelations (right inset) corresponding to the adjacent panels in d.

Regarding the fast response of our system, we did provide the perturbation study shown in Fig. 3 and also experimental results for fast variations of the amplifier current once the whole system was thermalized, which allowed us to better evaluate the constants governing the optical performance of the system. We added to the Methods:

The system presently reacts from the “cold” state in a few seconds, compatible with the bulk nature of our commercial amplifier.

b) The trajectory represented in Fig2a starts from a point in the high gain region. That does not correspond to actual start-up trajectories, initiated from a pump off. What happens when the pumping power ramp is varied? Does the trajectory enter the stable soliton region in Fig 2a earlier (at lower gain), followed by an adiabatic evolution? This seems in contradiction with the large fluctuations seen during the ramp in fig 2d. Or maybe Fig2a is just too schematic?

The Reviewer is correct - Fig.2 a is in fact a representative schematic only, and does not account for the details of the ramp-up of the gain from an “off” state. In essence, it represents a ramp up time with a duration of 0s. The large fluctuations, however, are there and this case represents, in some sense, the ‘worst case scenario’ for inducing the perturbations. In Fig. 2a each fluctuation induces an additional ‘loop’ in the attractor. This is better seen in Fig. S2 which we report below for completeness, where we use a colour scheme to follow the system in time. As shown in panel b, the intensity has strong peaks, and the system shuts down several times before reaching the single soliton state. We have added this figure also in the Extended Data set.

Figure E2 Start-up in the presence of gain-induced nonlinearity and thermal nonlinearity: single soliton formation. Numerical propagation for Eqs. (1-4), modelling a microresonator-filtered fibre laser with the inclusion of a saturable gain and gain-induced nonlinearity in the amplifying cavity, and a thermal nonlinearity in the microcavity. The system parameters are $\Gamma_T = 5$, $\tau_T = 8 \times 10^3$, $\eta = 0.4$, $\tau_g = 4 \times 10^4$, $\Theta = -13$, $g = 0.25$, and $\Delta + \Theta g_P = -0.21$. **a** Pseudo-colour map of the electric field intensity in the microcavity, in the normalised units of Eqs. (1-4), as a function of the position in the microresonator x (which is normalised against the microcavity roundtrips) and time t (which is normalised against the main-cavity roundtrips). **b** Temporal evolution of the peak intensity. The colours varying for increasing times matches with the plot inside panel e (showing the attractor). **c** Temporal evolution of the effective detuning $\Delta_b - \Delta_a$. The colours varying for increasing times matches with the plot in panel e. **d** Temporal evolution of the gain. The colours varying for increasing times matches with the plot in panel e. **e** Map of the attractor for the peak intensity, detuning and gain, as in panel c-d, following the colour code of panel b as a function of time. The attractor is superimposed to the soliton (orange) and zero state (blue) stability regions, as in Fig. 2a of the main text.

3) For some reason still unclear, the 2-soliton state does not yield a symmetric time distribution. Yet, the distribution seems to equalise with an increased pumping, as can be evidenced from figure 4a, upper autocorrelation trace. Can the authors elaborate on that?

The Reviewer is correct, the two-soliton states are not generally equally spaced, and the non-equalised autocorrelation indicates exactly this: these are two identical pulses which, however, are not spaced equidistantly within the microcavity. This is also confirmed by the modulated spectrum and is clearly visible in the additional example that we reported in Fig. 1i and k.

We stress, however, that multi-soliton states generally are not expected to be necessarily equidistant and mutually locked, while the opposite is generally true. Multi-soliton states are

generally non-interacting, unless a specific form of interaction can be elicited in the system (e.g., in the form of long-range interaction of the soliton tails [see Jae K. Jang, Miro Erkintalo, Stuart G. Murdoch & Stéphane Coen, “Ultraweak long-range interactions of solitons observed over astronomical distances,” *Nature Photonics* 7, 657–663 (2013)]). Specific types of perturbations need to be included to elicit such interactions and lock/freeze the multiple solitons at a specific relative distance. [Jae K. Jang, Miro Erkintalo, Stéphane Coen & Stuart G. Murdoch, “Temporal tweezing of light through the trapping and manipulation of temporal cavity solitons,” *Nature Communications* 6, 7370 (2015)]

In our experiments, we observed quite a random distribution of the distance between the pulses of the two soliton states, often also evolving as the pulses propagate. This is expected from purely localised states that are not subject to strong long-range interactions. While we often observed them, equidistant multi-soliton cases appear to be not particularly predictable, and we also cannot definitively conclude that the upper states tend to have equidistant distributions.

Again, this is the intrinsic nature of multi-soliton pulses, and since our main focus is on achieving repeatable and reliable operation, demonstrating our ability to control the single soliton state is even more critically important: this is the state where the distance of the pulses does not fluctuate because there is only a single pulse per microcavity roundtrip. We have added the comments below to clarify this point:

The spectra and autocorrelation of the two soliton states indicate that the spacing of these pulses within the microcavity are not generally equidistant. Amongst our extensive set of experimental data, we have observed a random distribution of the distances of the two soliton states, often evolving in time. This confirms the localised behaviour of these pulses.

4) Figure 3jk: why taking the RF of the fundamental frequency of the long cavity, whereas it is operated at high harmonics? To discuss stability of the pulsed regime, it would be more instructive to take the RF spectrum at 48.9 GHz. In the text, the authors indicate an ultra-low noise radio-frequency spectrum, but this is not really substantiated, with a SNR limited to around 20dB.

We thank the Reviewer for the suggestion. We focused on the slow instabilities because they usually dominate in highly harmonic fibre cavities, that are typically governed by supermode instabilities, so we took care to show that such instabilities were absent. We had a SNR at 20dB only for a very narrow component at 100 MHz representing this type of supermode instability noise. We took care to improve our results and in our repeated set of data this value is reduced to -60 dB, shown below.

This component is localised in the modes at 1538 nm, carrying a small portion of a coexisting supermode. We discussed this in the Supplementary and Fig.E1 (see above). Following the Reviewer’s suggestion, in the revised manuscript we have also included the RF spectrum at 50 GHz in a newly added section on the pulse characterisation - in the Methods. Note that a direct

measurement at 50 GHz is a bit beyond our equipment capabilities (we do not have high enough bandwidth electronics). Hence, we heterodyned the signal with an electro optical modulator and detected a high order harmonic. We have added in the Methods section what follows:

Finally, Fig. E1 g,h reports the RF characterisation around the repetition rate frequency. A small portion of the output through-port signal was processed with an electro-optic modulator, leading to additional sidebands around each of the original comb lines. Since the electro-optic modulator was driven in saturation with a GPS-referenced microwave oscillator, several harmonic sidebands were generated, whose frequency distance from the comb lines was a multiple of the modulating signal frequency¹⁶. We considered the third harmonic sidebands, and we set the modulation frequency such that the interaction between adjacent comb-lines produced a $f_0=500$ MHz beat-note. We revealed it with an amplified photodetector and analysed it with an Electrical Spectrum Analyser (ESA). Quite evidently, in all the ESA traces, the repetition rate beat-note SNR is more than 40 dB, being here limited mainly by the ESA noise floor.

And the figure

Figure E1...g Radio-frequency spectrum measured around the repetition rate frequency via heterodyne modulation. A small portion of the ‘through’ output signal is processed with an electro-optic modulator, leading to additional sidebands around each of the original comb lines. We considered the third harmonic sidebands, and then set the modulation frequency such that the interaction between adjacent comb-lines produced a $f_0=500$ MHz beat-note. Electrical spectrum analyser trace of the signal derived from the amplified photodiode after the electrooptic modulation, at 50 MHz span, and resolution bandwidth of 30 kHz. **h** Same as (g) but with 1 MHz span, and resolution bandwidth of 10 kHz.

5) From Fig 4c we see that the needed thermal redshift of the main laser cavity exceeds just a little that of the microcavity. Could the author explain better how this is not just good luck? Indeed, as the authors write, nonlinearities of the same focusing type nearly cancel each other. Would other materials for the nonlinear microcavity be suitable? How to design both cavities to routinely benefit from that situation? Is playing with lumped losses a practical or even a required solution (I acknowledge the important consideration about these losses in the suppl. material section)? It is important to better assess the practical reach of the authors’ findings in that respect.

This is a crucial point of our manuscript, and we thank the Reviewer for raising it. Regarding the role played by losses, the Reviewer is correct: the losses are influential in adapting the effective nonlinearity of the system because they enable balancing the energy ratio between the main cavity and the microcavity. There is another important point. The gain material (and in general *any* gain material) provides a practical degree of freedom to directly modify the effective, slow, nonlinear coefficient. This is a very fundamental and general mechanism because any resonant gain also changes the refractive index of the material (via the fundamental Kramers-Kronig relationship: the more the gain increases, the larger is the step of the refractive index value between the red and blue frequency around the resonance). This change of the refractive index value is then proportional to the gain. But the effective gain can be saturated by the optical field, which eventually changes the refractive index in proportion to the optical intensity, creating an ‘equivalent’ thermal nonlinearity. Remarkably, the sign of this thermal nonlinearity changes between the red and the blue sides of the resonance. The red sign of the resonance, where the refractive index decreases with gain, undergoes

an effective increment as the optical power saturates the gain, and hence this is a focusing nonlinearity. We are indeed operating in the red region of the EDFA resonance. **To this aim, the presence of a 12 nm intracavity filter is very important because allows us to operate in this region.**

We added the new section in the methods:

Dependence of the nonlinear gain refractive index on pump power.

Erbium amplifiers have a resonance around 1538 nm and display a strong, step like nonlinear dispersion which changes sign around the resonance and increase in magnitude with pump power until saturation. Specifically, the well-known spectral response of refractive index and gain of Erbium shows a resonant behaviour around 1538 nm, with the classical, step-like response of the refractive index ruled by Kramers and Kronig relations. In particular, the jump in the refractive index response increases with the magnitude of the gain, resulting in a decrease of the refractive index for wavelengths longer than the resonance and an increase for shorter wavelengths.

Notably, because the gain saturates with the circulating laser power within the fibre cavity, this relationship means that the refractive index dependence with circulating laser power is defocusing for wavelengths shorter than 1538 nm and focusing for longer. In our experiment, using the intracavity 12 nm filter, we select this portion of the spectral gain. Because the displacement of the refractive index directly depends on the gain, and hence on the pump power, the nonlinearity provided by the gain can be controlled with the 980 nm pump. Remarkably, the gain material (and in general *any* gain material) provides then a practical degree of freedom to directly modify the slow nonlinear response of the system.

The modelling reported in Supplementary section S1 and S3 exactly describes this behaviour and is obtained from the very general Maxwell-Bloch relationships which, practically, are the simplest approximation of any gain material. Hence any gain material can be, in principle, adapted for this purpose.

Our model exactly describes this behaviour and is obtained from the very general Maxwell-Bloch relationships which, in practice, are the simplest approximation of any gain material. Hence any gain material can be, in principle, adapted for this purpose. This is a powerful point since it means that our system is completely general and can be applied to any material system or platform that can provide optical gain.

Section S3 in the supplementary materials demonstrates the theoretical validity of this general behaviour: at low gain, the nonlinear gain is not saturated enough to play a role in counteracting the thermal nonlinearity of the ring and the system produces blue-detuned patterns. At high gain, however, the refractive index is strongly affected by the gain. When the gain saturates, the refractive index increases and creates this additional nonlinearity.

We believe that it is important to highlight this point in the main text and so in the conclusions we have added:

Moreover, our theoretical model shows that, by employing the very general Maxwell-Bloch equations, any common gain material can be used to tailor the nonlocal nonlinearity.

In conclusion, we believe that we have addressed both Reviewers' comments fully and that the paper is now at the level that can be published without further corrections.

Reviewer Reports on the Second Revision:

Referees' comments:

Referee #2 (Remarks to the Author):

The authors have carried out this second revision very professionally. Not only did they respond to all comments in detail in a convincing manner, backed up with new data where needed, but also took advantage of the interaction with reviewers to further mature and expand their explanations of the delicate physical mechanisms involved. In this way, the manuscript – for which I was already supportive after the first round of revision – has gained even more reach. Therefore, I now strongly support publication of the manuscript in Nature in its current form.

Referee #4 (Remarks to the Author):

I have carefully read the manuscript, all of its accompanying supplementary materials and the author's response to the reviewers. It appears overall the authors provided intelligent response to the reviewer's comments and most of the reviewer's questions have been addressed. Another question which I would like the authors to clarify is related to Fig. 3(e) and (g). It is apparent from Fig.3(a) that these data were recorded merely seconds apart (with a system disruption in-between of course) yet they are visibly different. These indicate to me that the oscillating soliton lines do not end up with the same detunings from one trial to next. Implication is of course that the pulse repetition rate may change. Moreover, the detuning profile cannot be fitted by a single straight line, meaning the spectral line spacing may not even be constant between neighboring pairs of lines but is varying irregularly across the spectrum by at least a few megahertz. The authors should provide an explanation for this and what the implication is for this source to function as a frequency comb, should the reason behind the observation be fundamental.

The technical completeness of the manuscript is impressive and commendable. The manuscript is informative and educational. It is clear the authors have thoroughly investigated their system and gained very good understanding of it. Yet, I am not quite sure if Nature is the best fit for this work. My impression from reading previous microcomb-related Nature articles is they provide feats or proof-of-concept demonstrations which show potential to be of broad interest beyond just optics/photonics. Examples are "Marin-Palomo et al Nature 546, 274-279 (2017)", "Spencer et al Nature 557, 81-85 (2018)", "Stern et al Nature 562, 401-405 (2018)", "Riemensberger et al Nature 581, 164-170 (2020)". It is not certain if the same can be said of this work. Furthermore, bulk components such as 12 nm wide band pass filter are necessary elements for the scheme to work. This raises a question if the scheme could be miniaturized to be competitive with integrated systems in, e.g "Stern et al Nature 562, 401-405 (2018)" or "Shen et al Nature 582, 365-369 (2020)". Finally, although the concepts of self-emergence and balance of slow nonlinearities are new and significant, these are still ultimately built on the same system the authors reported in "Bao et al Nature Photonics 13, 384-389 (2019)". That work already showed much of what is shown here like theoretical modeling (no slow nonlinear variables though) and laser spectroscopy of detunings. For these reasons, it seems to me that this work is more appropriate for more specialized journals such as Nature Photonics.

The above however, is admittedly a subjective viewpoint and the ultimate decision rests with the editors.

Author Rebuttals to Second Revision:

Referee #2

The authors have carried out this second revision very professionally. Not only did they respond to all comments in detail in a convincing manner, backed up with new data where needed, but also took advantage of the interaction with reviewers to further mature and expand their explanations of the delicate physical mechanisms involved. In this way, the manuscript – for which I was already supportive after the first round of revision – has gained even more reach. Therefore, I now strongly support publication of the manuscript in Nature in its current form.

We thank the Referee for their support and all their valuable comments in the three review rounds, which have been critical to improving the quality of our work.

Referee #4

I have carefully read the manuscript, all of its accompanying supplementary materials and the author's response to the reviewers. It appears overall the authors provided intelligent response to the Reviewer's comments and most of the Reviewer's questions have been addressed.

We are pleased with the Referee's support.

Another question which I would like the authors to clarify is related to Fig. 3(e) and (g). It is apparent from Fig.3(a) that these data were recorded merely seconds apart (with a system disruption in-between of course) yet they are visibly different. These indicate to me that the oscillating soliton lines do not end up with the same detunings from one trial to next. Implication is of course that the pulse repetition rate may change. Moreover, the detuning profile cannot be fitted by a single straight line, meaning the spectral line spacing may not even be constant between neighboring pairs of lines but is varying irregularly across the spectrum by at least a few megahertz. The authors should provide an explanation for this and what the implication is for this source to function as a frequency comb, should the reason behind the observation be fundamental.

This is a stimulating and important point and brings to some interesting discussion. In the manuscript, we had already pointed out that the difference between the two detuning sets is extremely small - within a few per cent of the microcavity linewidth. In fact, this is within our experimental error. We have now added clear error bars in the graph. In particular, while the detection of the lasing lines has accuracy below the MHz, which enabled the analysis in section S2.1, the detection of the microcavity resonance under lasing conditions is less accurate. In that sense, hence, this difference is not significant. To highlight this, we show the detunings of the two datasets below, superimposed, in the top plot. This clearly highlights the strong similarity between the two sets, with their difference, shown in the bottom graph as a percentage of the microcavity linewidth (120 MHz). Very importantly, this difference is randomly distributed, highlighting no significance of deviations below about 5% of the microcavity linewidth. Hence, the small measured difference between the two states is not significant and simply reflects the expected experimental error in genuine data. We have added clear error bars in Fig.3d,e and clarified this in the manuscript:

Figure 3a shows that the soliton state consistently reappears even after strong system disruptions induced by external perturbations (Fig. 3a). The spectra in Fig. 3b,c show how the same soliton state reliably recovers and how the comb lines return in the same position within the microcavity resonances, given our experimental accuracy (Fig. 3d,e and f,g).

Detuning from dataset in Fig. 3e,g for the soliton recovery. Top: detuning of the two sets, showing their strong similarity. Bottom: difference between the two detuning sets normalised against the microcavity linewidth, 120 MHz.

Regarding the detuning position deviating from a straight line, the Reviewer is simply observing the microcavity dispersion of the resonance frequencies. The distribution of the detuning does not represent the spacing between the oscillating modes. In fact, we recall that the detuning measures the difference between the comb lines and the microcavity centre in linear conditions. These quantities are extracted by laser scanning spectroscopy.

In our work, the microcomb laser lines themselves belong to a coherent soliton state and, as such, they are rigorously equally spaced. The coherence of the state is proved by the standard metrological measurements that we present here. Such characterisations include autocorrelation and radio-frequency noise measurements for all of our datasets and radio-frequency measurements of the repetition rates, shown in Fig. E1. We covered this matter in detail in the previous rounds of reviews, particularly regarding the single soliton nature of our pulses. The Reviewer has clearly endorsed our technical discussion, so we feel that it is unnecessary to repeat it here.

In contrast to the line spacings of the oscillating pulses, the microcavity physical resonances themselves are not equally spaced due to the natural dispersion of the structure. Directly answering the Reviewer's query, this has no consequence on the quality of our comb – it simply is a reflection of the microcavity dispersion.

The technical completeness of the manuscript is impressive and commendable. The manuscript is informative and educational. It is clear the authors have thoroughly investigated their system and gained very good understanding of it.

We thank the Reviewer for the support.

Yet, I am not quite sure if Nature is the best fit for this work. My impression from reading previous microcomb-related Nature articles is they provide feats or proof-of-concept demonstrations which show potential to be of broad interest beyond just optics/photonics. Examples are "Marin-Palomo et al Nature 546, 274-279 (2017)", "Spencer et al Nature 557, 81-85 (2018)", "Stern et al Nature 562, 401-405 (2018)", "Riemensberger et al Nature 581, 164-170 (2020)". It is not certain if the same can be said of this work. Furthermore, bulk components such as 12 nm wide band pass filter are necessary elements for the scheme to work. This raises a question if the scheme could be miniaturised to be competitive with integrated systems in, e.g "Stern et al Nature 562, 401-405 (2018)" or "Shen et al Nature 582, 365-369 (2020)".

We value the Reviewer's own views but we respectfully disagree with this assessment. Certainly, the papers listed by the reviewers are terrific achievements, particularly toward telecommunication and optical integration. Our work, however, represents a major

fundamental advance for microcombs, based on new, innovative and fundamental physics that finally resolves several open challenges that are key in the field and beyond it.

Indeed, here we show how to transform the soliton states into *dominant attractors* of the system using general principles of nonlinear physics, which are widely applicable in different settings. The practical implications for the field of microcombs are significant: we have finally demonstrated a set-and-forget system that always starts naturally into any desired state and, most importantly, self recovers to that same state even after total disruption. None of these features has been achieved before, and all of them are absolutely necessary to translate microcombs into real-world practical applications.

Moreover, from a fundamental physics viewpoint, we obtain a striking diagram of states for our microcombs, recalling how matter exhibits a fixed phase (gaseous, liquid or solid) in ranges of global parameters (temperature and pressure). Similarly, our microcomb always reaches the same type of state when a set of global parameters (optical gain and cavity length) are set and fixed.

Our results are transformative in the field of microcombs and pave the way to fully integrated solutions that will display the same level of performance that we achieve here. The solutions to the key challenges are universal – advances in integration may very well be the subject of future work.

Finally, although the concepts of self-emergence and balance of slow nonlinearities are new and significant, these are still ultimately built on the same system the authors reported in “Bao et al Nature Photonics 13, 384-389 (2019)”. That work already showed much of what is shown here like theoretical modeling (no slow nonlinear variables though) and laser spectroscopy of detunings. For these reasons, it seems to me that this work is more appropriate for more specialised journals such as Nature Photonics.

The above however, is admittedly a subjective viewpoint and the ultimate decision rests with the editors.

We respectfully disagree with this comment. We carried out a generational evolution of the system and its understanding. Concerning the modelling, the addition of two coupled, slow nonlinear equations is not trivial – it is major and fundamental and, more importantly, it profoundly changes the very nature of our interaction with the system. Regarding measuring the detunings, laser scanning spectroscopy techniques are popular in the field (see also the recent Power-efficient soliton microcombs, Óskar B. Helgason, Marcello Girardi, Zhichao Ye, Fuchuan Lei, Jochen Schröder, Victor Torres Company) and are now a standard measurement approach.

Generally speaking, the simpler the system, the more fundamental, innovative and universal are the demonstrated concepts. This is the case of our work. The very goal of the microcomb research field is to achieve high-performance operation with as few components as possible – ideally, simply with a nonlinear microcavity and a pump laser. The general bottom line of microcomb research is – the simpler the system, the better. Again, our work demonstrates the universality of a new and fundamental type of microcomb physics, and we present a clear and comprehensive theoretical and experimental understanding of this here.

Finally, we ultimately appreciate the intellectual honesty of the Reviewer, pointing out that this criticism is a question of personal taste. We accept this but at the same time provide evidence of the breakthrough, novelty, relevance, usefulness, impact and broad interest of our results, thus justifying their presentation on the pages of Nature.

nature portfolio